# Numerical model derived intensity-duration thresholds for early warning of rainfall-induced debris flows in a Himalayan catchment

Sudhanshu Dixit[1,*], Srikrishnan Siva Subramanian[1,*@], Piyush Srivastava[1], Ali. P. Yunus[2], Tapas Ranjan Martha[3], and Sumit Sen[1]

[1]Centre of Excellence in Disaster Mitigation and Management, Indian Institute of Technology Roorkee, Roorkee, Uttarakhand-247667, India
[2]Department of Earth and Environmental Sciences, Indian Institute of Science Education and Research Mohali, Punjab-140306, India
[3]National Remote Sensing Centre (NRSC), Indian Space Research Organisation (ISRO), Dept. of Space, Govt. Of India, Balanagar, Hyderabad - 500037, India
[*]These authors contributed equally to this work.
**Correspondence:** @ Srikrishnan Siva Subramanian (srikrishnan@dm.iitr.ac.in)

**Abstract.** Debris flows triggered by rainfall are catastrophic geohazards that occur compound during extreme events. Few early warning systems for shallow landslides and debris flows at the territorial-scale use thresholds of rainfall Intensity-Duration (ID). ID thresholds are mostly defined using hourly rainfall. Due to instrumental and operational challenges, current early warning systems have difficulty forecasting sub-daily time series of weather for landslides in the Himalayas. Here, we present

a framework that employs a spatio-temporal numerical model preceded by the weather research and forecast (WRF) model for analysing debris flows induced by rainfall. The WRF model runs at 1.8 km × 1.8 km resolution to produce hourly rainfall. The hourly rainfall is then used as an input boundary condition in the spatio-temporal numerical model for debris flows. The debris flow model is an updated version of Van Asch et al. (2014) in which sensitivity to volumetric water content, moisture content-dependent hydraulic conductivity, and seepage routines are introduced within the governing equations. The spatiotemporal

numerical model of debris flows is first calibrated for the mass movements in the Kedarnath catchment that occurred during the 2013 North India Floods. Various precipitation intensities based on the glossary of the India Meteorological Department (IMD) are set and parametric numerical simulations are run identifying ID thresholds of debris flows. Our findings suggest that the WRF model combined with the debris flow numerical model shall be used to establish ID thresholds in Territorial Landslide Early Warning Systems (Te-LEWS).

## 1 Introduction

Rainfall-induced debris flow disasters are catastrophic and affect people's livelihood in mountainous regions (Cannon and DeGraff, 2009; Stoffel et al., 2014; Turkington et al., 2016). The increasing frequency and number of extreme-rainfall events driven by climate change may aggravate the occurrence of disastrous debris flows in several regions around the world (Field et al., 2012; Dash and Maity, 2021; Westra et al., 2014; Bharti et al., 2016). These make debris flow disaster mitigation

an urgent need (Suzuki et al., 2020). Structural and non-structural mitigation measures are practised to mitigate debris flows

impacts (Fan et al., 2019; Huebl and Fiebiger, 2005). However, non-structural mitigation measures, i.e., early warning systems, adapt to practices of efficient early warning and are implemented on larger scales, which is essentially required during extreme events (Piciullo et al., 2018; Guzzetti et al., 2020). Nations, i.e., the United States of America, Japan, Italy, and China, have developed debris flow early warning systems that work at the territorial scale, few of them covering certain regions and other few an entire nation (Baum and Godt, 2010; Osanai et al., 2010; Ju et al., 2020; Alfieri et al., 2012). These systems use radar-based rainfall forecasts and observed data derived Intensity-Duration (ID) of rainfall to set the triggering thresholds of landslides for early warning (Bogaard and Greco, 2018; Brunetti et al., 2010; Guzzetti et al., 2008; Staley et al., 2013). With the help of historical records of debris flows and their corresponding triggering rainfall intensity and duration, the determination of thresholds is usually considered in these early warning systems (Intrieri et al., 2013; Segoni et al., 2018; Stähli et al., 2015).

In India, the National Remote Sensing Centre (NRSC) under the Indian Space Research Organisation (ISRO), developed an "Experimental Landslide Early Warning System for Rainfall Triggered Landslides" along selected road corridors in Uttarakhand, India (Jayaraman, 2013; Khatri et al., 2017; Bharwad, 2019). The system's historical landslide data and rainfall records are sourced from the Border Roads Organization (BRO) and the India Meteorological Department (IMD). Experimentally forecasted rainfall data from the Space Applications Centre (SAC) and landslide hazard zonation maps from NRSC are used. Combinations of 24-Hour rainfall and various antecedent durations based thresholds are statistically combined to estimate the probability of landslide occurrences (Mathew et al., 2014). The thresholds used were determined based on 3 hourly rainfalls from TRMM 3B42 V.6 data. Whereas the actual LEWS operated by National Remote Sensing Centre (NRSC), Indian Space Research Organisation (ISRO) uses daily as well as multiple days antecedent rainfall based on Mathew et al. (2014). However, the 24-hourly/daily rainfall threshold may perform well for predicting shallow landslides but not for debris flows. Runoff generation depends on shorter duration rainfall intensities, and early warning system thresholds of hourly rainfall become fundamental for debris flows (Pan et al., 2018; Hürlimann et al., 2019). Due to instrumental and operational challenges, current early warning systems have difficulty forecasting sub-daily time series of weather for landslides in the Himalayas (Bhandari, 1987, 2006; Gariano et al., 2023).

Many nations use Territorial Landslide Early Warning Systems (Te-LEWS) as a cost-effective non-structural mitigation measure for landslides (Piciullo et al., 2018; Guzzetti et al., 2020). However, most Te-LEWSs or models, i.e., ID, antecedent rainfall, or Soil Water Index (SWI), have genetic inaccuracies since traditional methods derive thresholds from statistical/data-driven correlations of past events and monitoring data (Lagomarsino et al., 2013). Implementing Te-LEWSs in new geological settings, i.e., the Himalayas and the Western Ghats, India, with limited historical events and precipitation records, is very challenging (Gariano et al., 2023). With a limited amount of recorded historical landslides, capturing the exact value of the threshold is cumbersome. An alternative approach is required to simulate landslides' occurrence under various magnitudes of precipitation and inform about the landslide-triggering conditions. Early warning systems that use ID for debris flows rely on an hourly forecast of rainfall data (Hürlimann et al., 2019). In India, however, the current thresholds are based on daily rainfall. The above reasons invite improvements to India's existing Te-LEWS.

In this study, we present a framework for an early warning system comprised of a weather research and forecast (WRF) model (Srivastava et al., 2022) followed by a spatiotemporal numerical model for debris flows (Van Asch et al., 2014; Domènech

et al., 2019; Siva Subramanian et al., 2021). Using the framework, we analyze the debris flows in Kedarnath, Uttarakhand, India, during the 2013 North India Floods. The hourly precipitation time series is obtained from the WRF simulations and compared with observations from the India Meteorological Department (IMD). Then, the triggering intensity-duration (ID) thresholds are derived through parametric numerical simulations under various rainfall intensities. As the study is introduced above, we organise the rest of the manuscript as follows. Section 2 introduces the debris flow event that occurred in Kedarnath, Uttarakhand, India, during the 2013 North India Floods. The data and methods adopted in this study are first detailed in Section 3. Then, the methodology is detailed, starting from the WRF model followed by the numerical model for debris flows (Van Asch et al., 2014; Domènech et al., 2019; Siva Subramanian et al., 2021). After this, the ID threshold method adapted in this study are presented briefly. The results of the numerical modelling and ID threshold analysis are presented in Section 4. In Section 5, we discuss the importance of hourly rainfall data for the early warning of debris flows in Uttarakhand, India, and highlight the improvements further needed to improve the precision of ID thresholds.

## 2 Study area and characteristics of the disaster

This study considers the 2013 extreme rainfall-induced debris flows in Kedarnath as a case example. The study area is located inside the Himalayan tectonic zone, and the landscapes here are weakened geological formations with undulated terrain, narrow valleys, and steep slopes (Fig. 1). The area is situated towards the north portion of the Main Central Thrust (MCT) and Tethyan Detachment Fault bounds the other direction. The rocks are composed of Higher Himalayan Crystallines of metamorphic origin with occasional granitic intrusions. Gneiss, Kyanite, Calc Silicate, Biotite Granite, Quartzite, Marble, and Schists are major rock types in this area (Fig. 2a). The unmapped area mainly surrounds the Glacier. Over this fragile terrain, due to extreme rainfall, over 120 landslides, mostly of debris flows and slides followed by flash floods, occurred during 15-17 June 2013 (Martha et al., 2013; Champati Ray et al., 2016; Allen et al., 2016). Martha et al. (2013) mapped a total of 6013 landslides over the entire Uttarakhand and found that 3472 landslides newly occurred over an area of 30.4 km$^2$. The disaster caused more than 5000 casualties and severe economic impacts. Surrounding the Kedarnath, India Meteorological Department (IMD) observed unprecedented extreme rainfall amounts of over 350 mm between 14 and 18 June 2013 (Dobhal et al., 2013; Singh et al., 2015). Numerical weather prediction model studies have also found the cumulative daily rainfall during 16 and 17 June were close to 200 mm (Shekhar et al., 2015; Kumar et al., 2016; Chevuturi and Dimri, 2016; Dube et al., 2014). Continuous rainfall occurred on 15, 16, and 17 June, triggering catastrophic debris flows through runoff-induced erosion of weak sediments overlying the hillslopes (see the location of landslides, mostly debris flows/slides in Fig. 1). Martha et al. (2013) mapped a total of 121 landslides within the Kedarnath catchment (Fig. 2b). Photographs taken during field work at Kedarnath Valley during December 2022 is shown in Fig. 3.

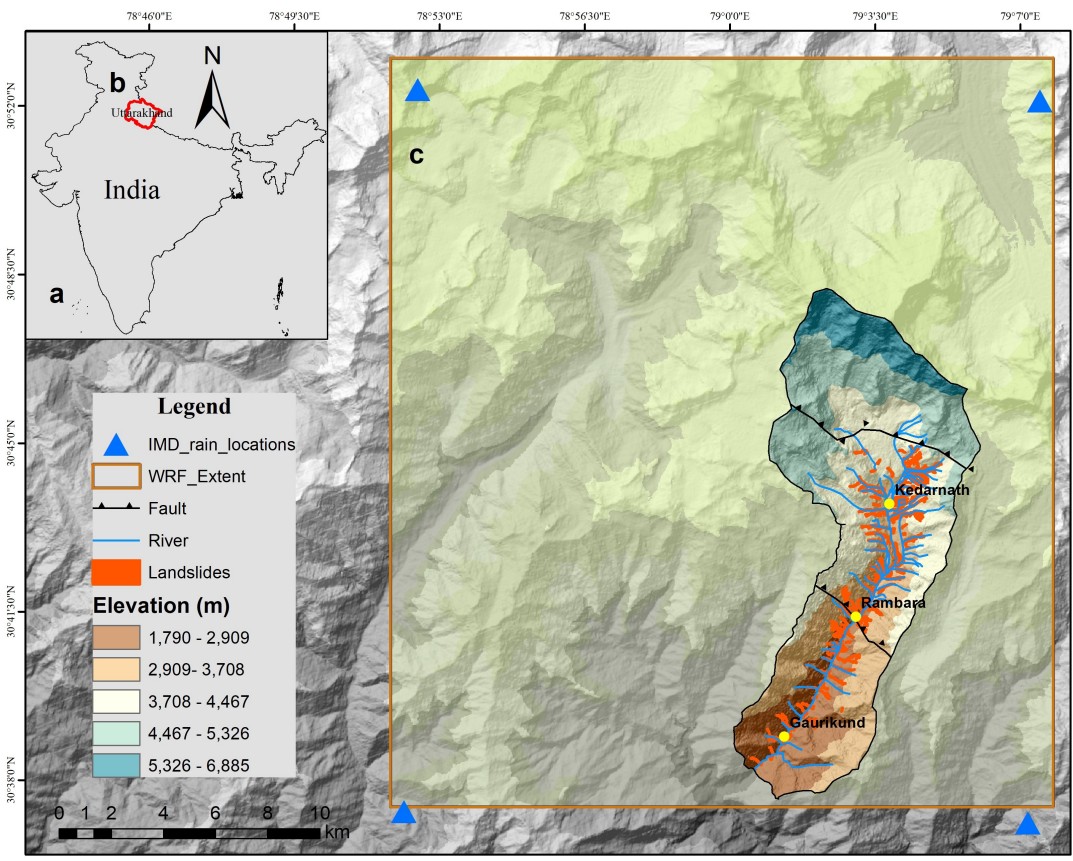

**Figure 1.** Map showing the extent of the study area within Uttarakhand, India. (a) India administrative boundary highlighting Uttarakhand (Copyright: Geological Survey of India, downloaded from Bhukosh), (b) the Location of Uttarakhand (Copyright: Geological Survey of India, downloaded from Bhukosh), and (c) The extent of debris flows overlaid by ALOSPalSAR 12.5m digital elevation model in and around the disaster sites in Kedarnath, Uttarakhand, India.The domain to retrive output from the Weather Research and Forecast (WRF) numerical model is shown in red box, the locations for data inferred from the India Meteorological Department (IMD) gridded data are marked in blue triangles, the landslides are shown in red Polygons. Two major faults run through the study area. The catchments possess 0, 1$^{st}$ and 2$^{nd}$ order streams.

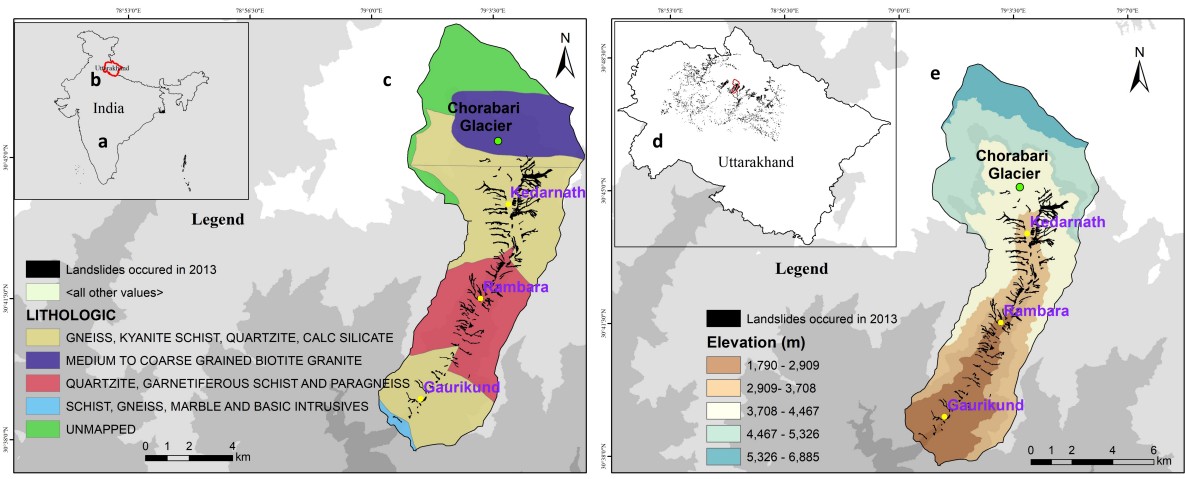

**Figure 2.** (a) India, (b) Uttarakhand, (c) Lithology of the Kedarnath catchment (Copyright: Geological Survey of India, downloaded from Bhukosh Portal, (d) All landslides within Uttarakhand state occurred during the 2013 North India Floods, (e) Extent of landslides that occurred from 15 to 17 June 2013, mapped by Martha et al. (2013).

## 3  Data and Methods

### 3.1  Meteorological data and WRF simulations

The methodological flowchart used for the modelling is shown in Fig.(4). The numerical modelling approach starts with the rainfall simulation in an hourly timestep. For this purpose, the Weather Research and Forecasting (WRF) numerical model version 4.2.2 (Skamarock et al., 2019) is used in this study. The WRF Model is a state of the art mesoscale numerical weather prediction system designed for atmospheric research and operational forecasting applications. In this study, this model is used for deriving the hourly rainfall timeseries during the 2013 North India Floods over Kedarnath region, Uttarakhand, India. (Srivastava et al., 2022) The model has a fully compressible setup with a non-hydrostatic dynamical core. The model uses terrain-following hydrostatic pressure over the vertical coordinates for numerical simulations. Fig.5 shows the geographical coverage of the WRF model setup. The black rectangular boxes represent two one-way nested domains, domain d01 (9 × 9 km) and domain d02 (1.8 × 1.8 km). The output for the study area is obtained by defining a rectangular box shown in Fig.1.

For comparison of the WRF derived rainfall, we use the India Meteorological Department (IMD) gridded data and Global Satellite Mapping of Precipitation (GSMaP) data respectively at daily and hourly rain rates with respective spatial resolutions 0.25 x 0.25 degree 0.1 x 0.1 degree. The meteorological data from IMD are obtained at Locations 1, 2, 3, and 4 (see Fig. 1, blue triangles). A comparison of the data from IMD with the WRF derived data (cumulative to daily timesteps) is shown in Fig.6 (a and b). The data from IMD comprises daily gridded rainfall information for India, featuring a spatial resolution of 0.25 x 0.25 degrees and spanning a lengthy period from 1901 to 2022. This dataset, covering 122 years, provides detailed insights into daily rainfall in millimeters. For the WRF simulation, ERA5 reanalysis data is used as initial boundary data. The temporal

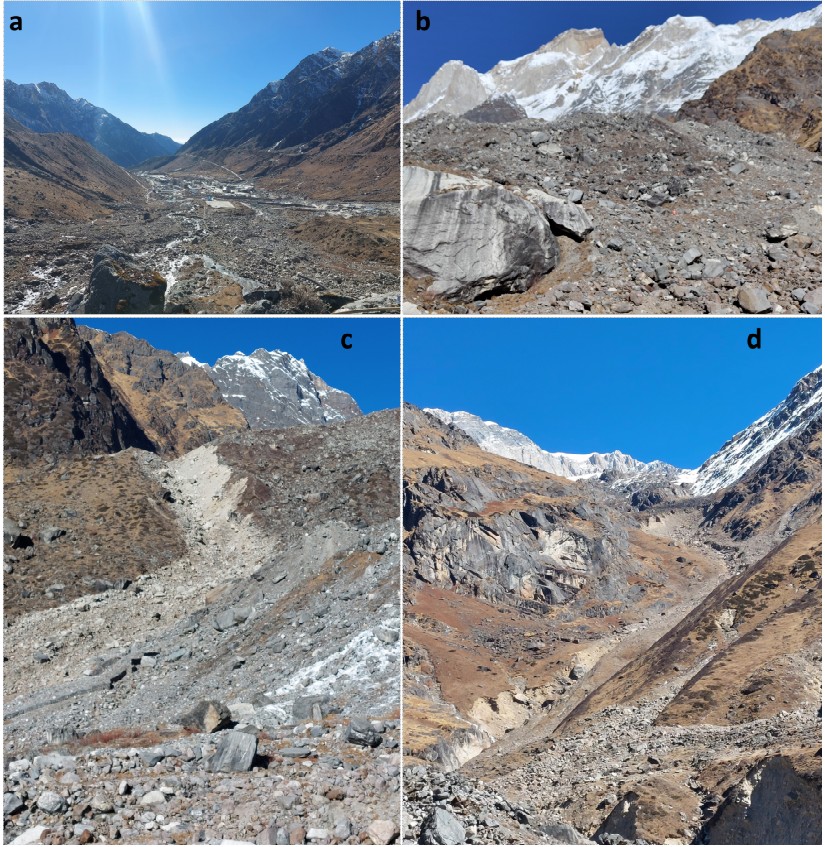

**Figure 3.** Photographs taken during field work at Kedarnath valley during December 2022. (a) Viewing from North towards downstream side of the Kedarnath valley, (b) Debris flow deposits approximately 1 km behind the Kedarnath Shrine, (c) Runout path of debris flow flood, (d) Major debris flow that hit the Chorabari glacier lake (this photograph was taken climbing above the debris flow deposits shown in Fig. 3(b)

interval of the ERA5 boundary data used is 6 hours. Further configuration of the WRF model to reproduce the analysis is shown in Fig.5.

Spatially explicit rainfall timeseries maps at an interval of 1 hour are derived as an output from the WRF model. The area used to derive the rainfall maps is shown in Fig.1. The WRF numerical model-based rainfall during the days 15 June, 16 June, and 17 June 2013 sourced precisely at the centre of 121 landslide polygons are plotted in Fig.7 (a,b and c) respectively for average, minimum, and maximum rainfall. The hourly rainfall timeseries from WRF for average, minimum, and maximum rainfall is compared with GSMaP data as shown in Fig.7 (a,b ,and c).

**3.2   Numerical modelling of debris flows**

In this study, we use the numerical model developed by Van Asch et al. (2014) and Siva Subramanian et al. (2021). The model is an updated version of Van Asch et al. (2014) in which sensitivity of soil moisture, moisture content dependent hydraulic

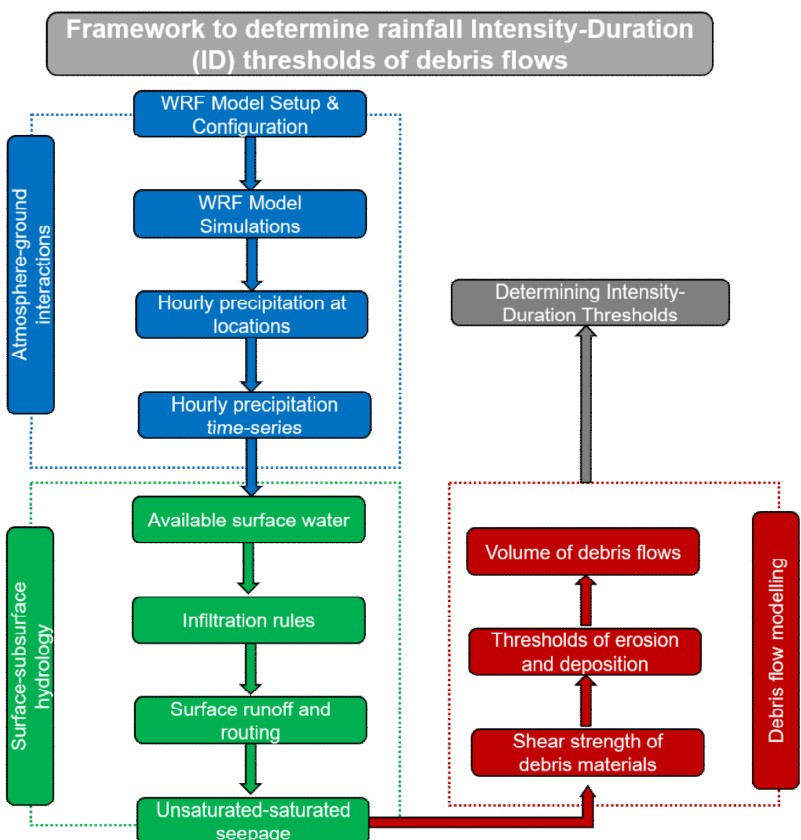

**Figure 4.** Methodological framework proposed in this study to determine the rainfall ID thresholds of debris flows

conductivity and seepage routines are embedded (Siva Subramanian et al., 2021). A Python-based script and command line are used to code the model in PcRaster, a dynamic programming tool based on a Geographical Information System (GIS) platform (Deursen, 1995). A digital elevation model (DEM) is required to run the model. The resolution of the DEM sets the mesh size of the model. In addition to the DEM, other DEM derivatives ,i.e., slope, and drainage direction, are input as maps having the same resolution. We use the publicly available 12.5m resolution ALOS PALSAR DEM for this study. Other spatial inputs, i.e., the area of the catchment, depth of soil or regolith (Hengl et al., 2017), area of precipitation, and Local Drainage Direction (LDD) maps are used. Pre-processing of these datasets is done using ArcGIS version 10.8.2 (Ormsby et al., 2004) readers are directed to Van Asch et al. (2014) and Van Asch et al. (2018) for more information on the source model's governing equations. The infiltration and seepage schemes are based on Siva Subramanian et al. (2020)'s equations. The complete governing equations of the model are available from Siva Subramanian et al. (2021) and are explained briefly below and detailed in the supplementary file.

In the model, debris flow initiation occurs when the bed shear stress ($\tau, kPa$) is larger than the critical erosive shear stress ($\tau_c, kPa$), and the volumetric concentration of solids in the debris flow ($C_v$) is smaller than an equilibrium value ($C_{V\infty}$). The

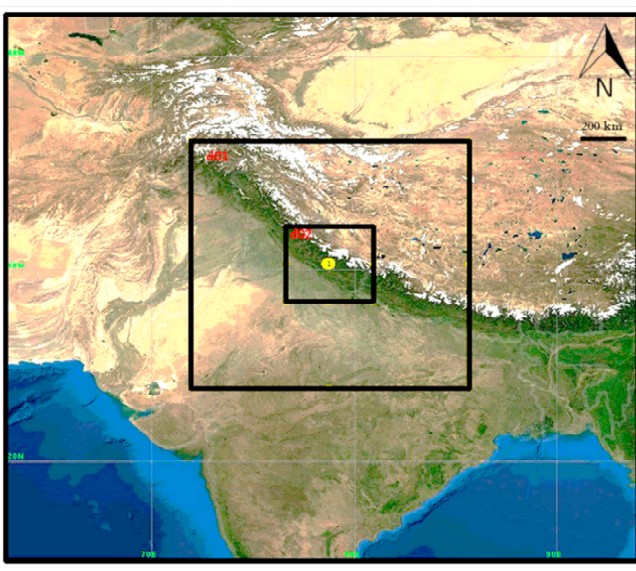

| Parameter | Description |
|---|---|
| Initial and boundary data | ERA5 reanalysis |
| Temporal interval of boundary data | 6 h |
| Grid size | Domain 1: (146*151) * 50<br>Domain 2: (226*231) * 50 |
| Horizontal resolution | Domain 1: 9 km<br>Domain 2: 1.8 km |
| Nesting | One way |
| Vertical levels | 50 |
| Time step | 15 s |
| Land Use and Land Cover | USGS data updates using AWiFS |
| Topographic data | GMTED2010 |
| Microphysics | Thompson scheme |
| PBL scheme | YSU scheme |
| Cumulus parameterization | Kain-Fritsch scheme |
| Shortwave radiation | Dudhia scheme |
| Longwave radiation | RRTM scheme |
| Land surface model | Noah-MP land surface model |
| Surface-layer | Revised MM5 scheme |

The two nested domains, namely domain d01 (9 km × 9 km) and domain d02 (1.8 km × 1.8 km) for WRF model simulations are shown with black rectangular boxes

**Figure 5.** The setup and configuration of Weather Research and Forecast (WRF) numerical model to derive hourly rainfall time-series from 15 to 17 June 2013. Modified from Srivastava et al. (2022)

equilibrium value, also referred to as the transport capacity of the flow, is defined based on stability theory using the expression proposed by Takahashi et al. (1992):

$$C_{V\infty} = \frac{\rho_w \tan\theta}{(\rho_s - \rho_w)(\tan\phi_{bed} - \tan\theta)} \tag{1}$$

where $\rho_w$ (kg/m³) is the density of water, usually assumed to be 1000 kg/m³, $\rho_s$ (kg/m³) is the density of the solids, $\phi_{bed}$ (°)

is the angle of internal friction of the bed/slope materials and $\alpha$ (°) is the slope angle of the hillslope derived from the DEM. The rate of erosion ($e_r$) is expressed based on Takahashi et al. (1992):

$$e_r = \delta_e \frac{a_c}{d_L} U = \delta_e \frac{C_{V\infty} - C_V}{C_{V*} - C_{V\infty}} \frac{q_t}{d_L} \tag{2}$$

where $\delta_e$ is the coefficient of erosion rate, which is non-dimensional and back-calculated for any given analysis, $a_c$ (m) is the depth within the sediment layer under the condition $\tau_c = \tau$ , $d_L$ is $d_{50}$ mean diameter of the grain, U (m/s) is the velocity of

the flow-through vertical section, $C_{v*}$ is the volumetric fraction of solids and $q_t$ (m²/s) is the routed total discharge of the sum

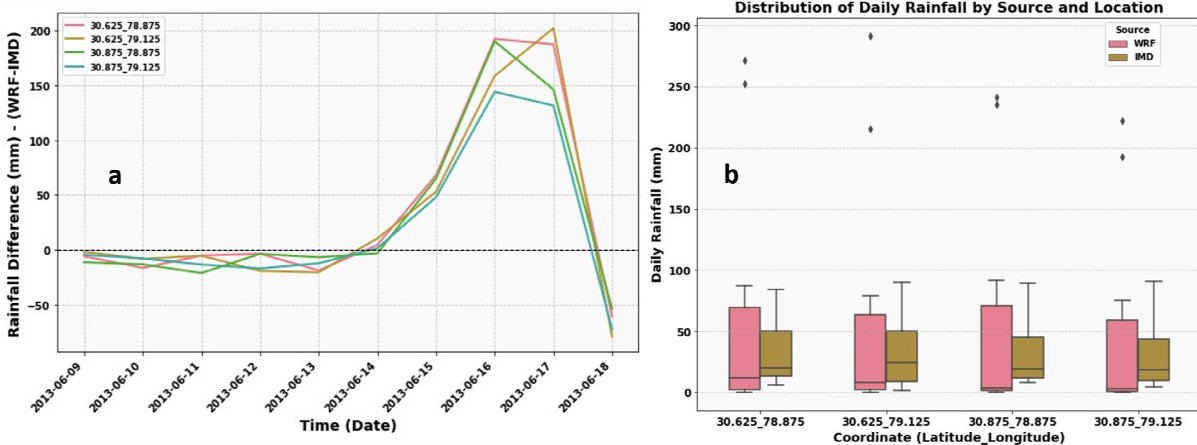

**Figure 6.** (a) Daily rainfall data at four locations around Kedarnath catchment (see Fig. 1), and (b) Distribution of daily rainfall at four locations from 15 to 17 June 2013 compared between WRF numerical model and IMD data

of sediments and water per unit width expressed by Van Asch et al. (2014). For further details of the governing equations of the source model, readers are referred to Van Asch et al. (2014) and Van Asch et al. (2018). Part of the infiltration and seepage schemes are from Siva Subramanian et al. (2020). The percolation-based infiltration model is available from Siva Subramanian et al. (2020) and van Beek (2002).

At first, the input parameters listed in Table 1 and the rainfall data acquired from the WRF model are given to the numerical model. Then, we run the numerical model (see Fig. 6). The numerical analysis will last for a total of three days, from June 15 to June 17, 2013. For the purposes of convergence, the time step is set to 1 hour and the total duration is 72 hours. Courant-Friedrichs-Lewy (CFL) condition checks the mass balance and convergence at every timestep (De Moura and Kubrusly, 2013). The model monitors changes in erosion and deposition, as well as volumetric water content response, at various catchment

areas. The volume of eroded debris is monitored at the confluence of first order and second order streams to the main river.

**Table 1.** Parameters used for the numerical analysis. $\rho_s$, $Cv*$, $\phi_b$, $\tau_c$, ks, $\mu$, and n are set referring to the literature (Siva Subramanian et al., 2021). d50, $\delta_e$, and $\delta_d$ are set by calibration and back analysis.

| Parameter | d50 (mm) | $\rho_w(kg/m^3)$ | $\rho_s(kg/m^3)$ | $Cv*$ | $\phi_b(^0)$ | $\tau_c$ | $\delta_e$ | $\delta_d$ | ks $(m/hr.)$ | $\mu$ | n |
|---|---|---|---|---|---|---|---|---|---|---|---|
| Value | 46.9 | 1000 | 2600 | 0.65 | 35 | 1 | 0.01 | 0.001 | 0.01565 | 1 | 0.04 |

In Table 1, d50 = mean grain size; $\rho_w$ = density of water; $\rho_s$ = density of solid particles; $Cv*$=volume fraction of solids in the erodible bed; $\phi_b$ = friction angle of soil; $\tau_c$ = yield strength; $\delta_e$ = coefficient of erosion rate; $\delta_d$ = coefficient of deposition rate; ks = soil infiltration capacity; $\mu$ = dynamic viscosity; n = Manning's number.

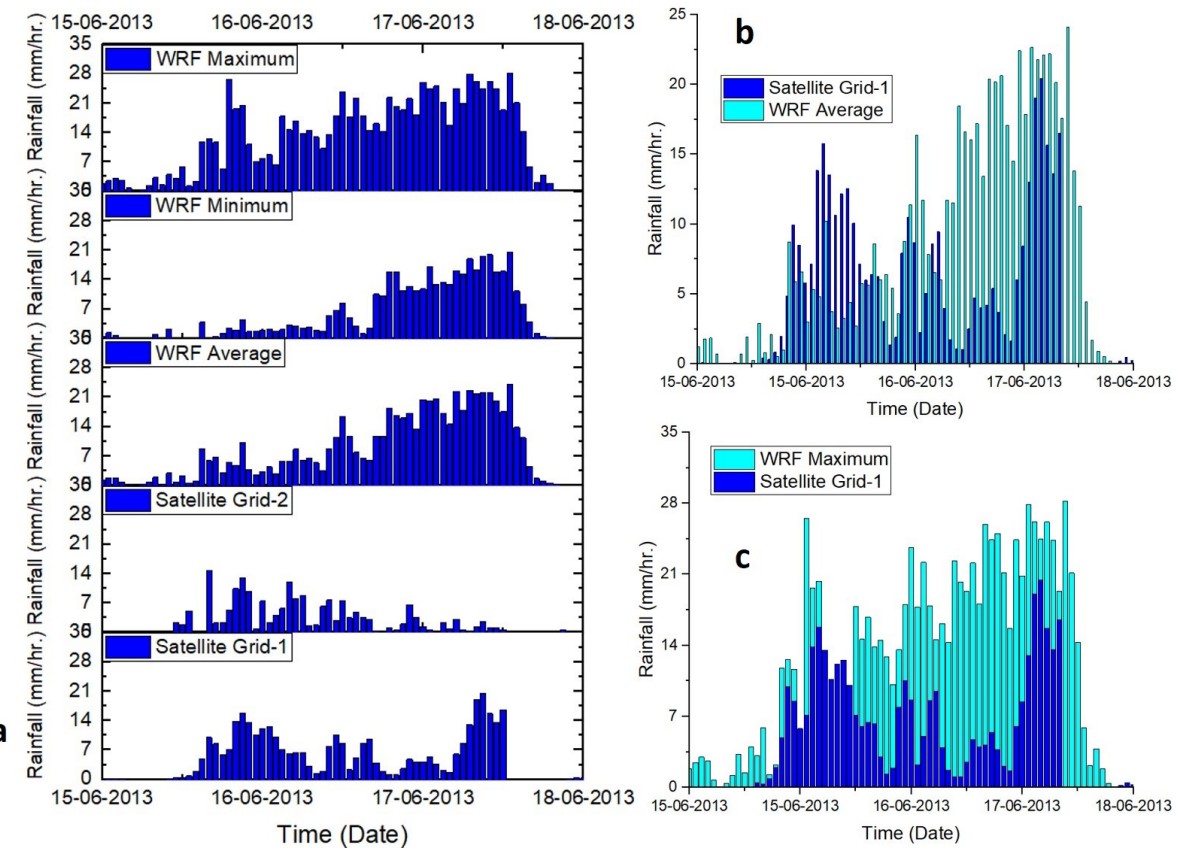

**Figure 7.** (a) Hourly rainfall time-series from 15 to 17 June 2013 derived from WRF numerical model: average, minimum and maximum over 121 locations of landslides within the study area. Satellite Grid-1 and Grid-2 shows the data from GsMap. (b) Average of WRF rainfall compared with Satellite data for region Grid-1, and (c) Maximum of WRF rainfall compared with Satellite data for region Grid-1

### 3.3 Determination of Intensity-Duration (ID) thresholds that trigger debris flows

Caine (1980) proposed the relationship between I (rainfall intensity) and D (rainfall duration) shown in Eq.3, which is now commonly used to establish rainfall thresholds in territorial landslide early warning systems (Te-LEWS) for shallow landslides and debris flows (Destro et al., 2017; Iadanza et al., 2016; Intrieri et al., 2013; Peruccacci et al., 2017; Scheevel et al., 2017).

$$I = \alpha D^{-\beta} \tag{3}$$

Here, two constant fitting parameters $\alpha$ and $\beta$ are used.

Traditionally, ID thresholds are determined statistically, correlating the rainfall data with the occurrence of landslides. A relationship proposed by Caine (1980) is plotted in Fig.9. In India, few studies have identified ID threshold for landslides in the Himalayas (Kanungo and Sharma, 2014; Mathew et al., 2014). Fig. 9 also shows a few proposed thresholds lines from

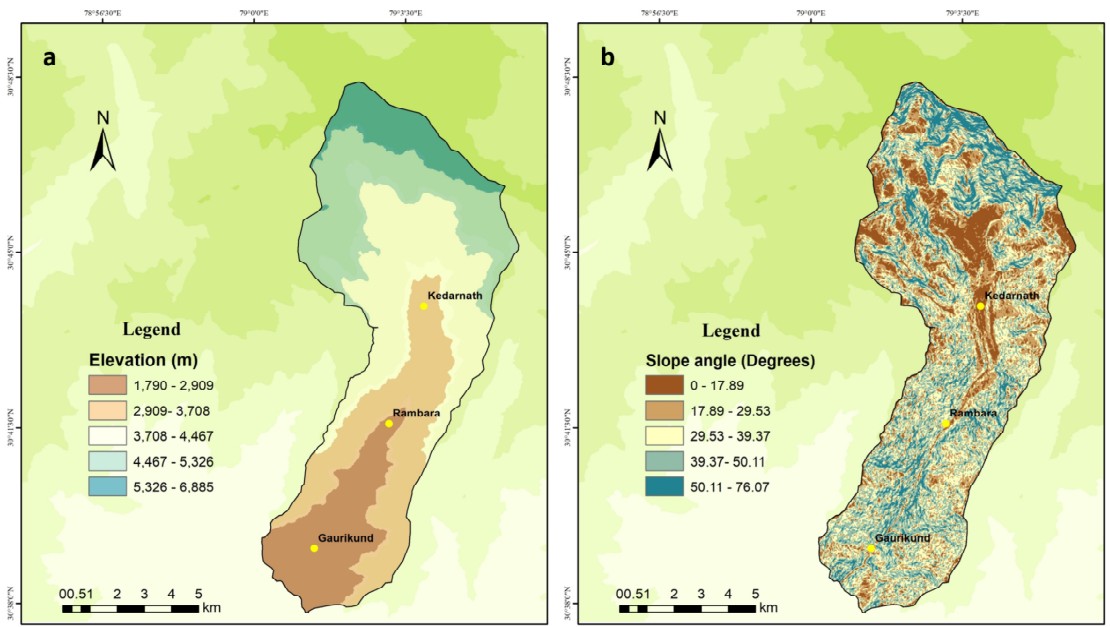

**Figure 8.** Spatial datasets required for numerical modelling. (a) Elevation (m), and (b) Slope angle (Degrees). Other spatial inputs for the debris flow model are shown in the Supplementary.

India. Very recently, Jiang et al. (2021) defined inter-event-time (IET) of rainfall episodes for debris flow early warning. Because of the global application of this approach, the same method is used in ISRO's experimental landslide early warning system (Mathew et al., 2014).

Since it is proven that these statistical thresholds hold a physical explanation of the initiation processes of debris flows (D'Odorico et al., 2005; Thomas et al., 2018; Berti et al., 2020), a numerical model that simulates debris flow triggering through rainfall and runoff-erosion shall be used to determine the ID thresholds. Where historical data is unavailable, these numerical simulations may be the best alternative to determine the triggering rainfall thresholds (Van Asch et al., 2014, 2018). At first, we calibrate the numerical model using the methods described above. By doing the calibration, we obtain the best suited values for the model's parameters, shown in Table 1. At this stage, the numerical model is ready to simulate the triggering of debris flows for any precipitation intensity. Instead of determining the ID thresholds using a statistical approach, we run the numerical model with different rainfall intensities based on IMD glossary (see Fig. S9 and Fig. S10 in the supplementary) in this study. Then we estimate the duration at which a debris flow will be triggered for the area where the model is already calibrated. Then, we run ten numerical simulations under rainfall intensities (I) ranging from 10 mm/hr. to 15 mm/hr., 20 mm/hr., 25 mm/hr., 30 mm/hr., 35 mm/hr., 40 mm/hr., 45 mm/hr., 50 mm/hr., 55 mm/hr., and 60 mm/hr. The duration (D) for each set of numerical simulations is observed tracking the arrival time of debris flow at the confluence. By correlating the intensity and duration derived from each set of numerical simulations, ID thresholds are established.

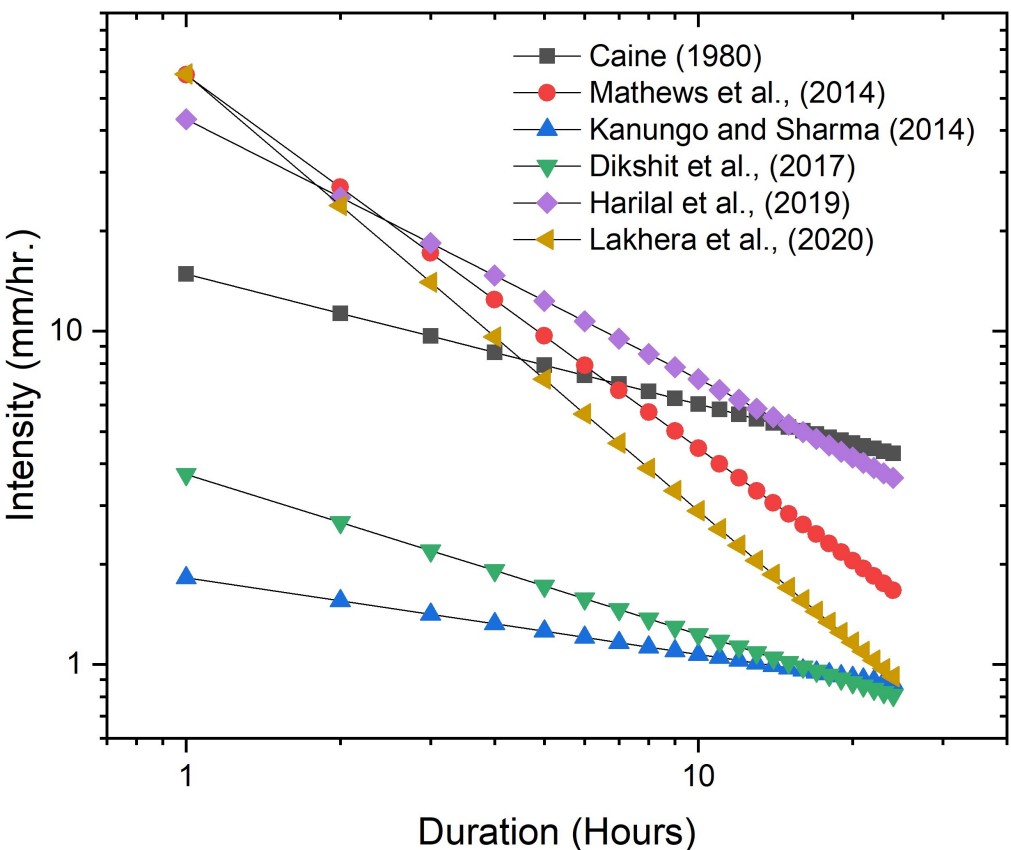

**Figure 9.** Intensity-duration rainfall thresholds used for classifying occurrences and non-occurrences of landslides for early warning. The plot shows ID thresholds determined by various studies (Caine, 1980; Mathew et al., 2014; Dikshit et al., 2020; Lakhera et al., 2020; Kanungo and Sharma, 2014; Harilal et al., 2019).

### 3.4 Estimating volume of debris flows

To calibrate the numerical model, estimating the volume of debris flow is essential. Martha et al. (2013) have mapped the landslide polygons of 2013 Kedarnath debris flows and have estimated the area of landslides. However, no information on the volume of debris flow is available from the literature. In this study, we use an empirical formulation of debris flow volume to estimate the total volume of sediments generated during the 2013 extreme rainfall event in the Kedarnath catchment. For this, the parameters i.e., area of the watershed, area of landslides, cumulative rainfall, and duration of rainfall are considered

following the empirical equation by Chang et al. (2011).

$$V = 0.023A_W + 0.064A_L + 13264.6GI - 1399.2D + 38.47C_R \tag{4}$$

Here, $A_W$ is the area of the catchment 94724450 (m$^2$), $A_L$ is the area of the landslides 2303777 m$^2$, GI is the Geological Index based on lithology is 5 (Chang et al., 2011), $D$ is the duration of rainfall is 72 hours, and $C_R$ is the cumulative rainfall

for the debris flow event that is 646 mm. The total volume of debris flows generated during the 2013 extreme rainfall within the Kedarnath catchment is estimated as 2320000 m$^3$. Here, it is not straightforward to expect an empirical equation originally developed for Taiwan to approximate debris flow volume in the Himalayas reasonably. However, the empirical equation proposed by Chang et al. (2011) accounts for different lithology, rainfall intensity, duration, and catchment characteristics ,i.e., Area of the watershed to estimate the volume of debris flows generated during extreme rainfall events. As discussed by Chang et al. (2011), the equation shall be applied to other areas having debris flows generated by extreme rainfall. In addition, Marchi and D'Agostino (2004) suggests that the debris flow sediment volume does not possess a strong sensitivity to the lithology. Rickenmann and Koschni (2010) suggest that the volume of sediments during debris flows is strongly affected by the flood volume i.e., runoff and catchment characteristics. Considering these advantages, in this study, the debris flow volume is estimated using the empirical equation given by Chang et al. (2011), and it seemed appropriate to use in the Himalayas.

## 4 Results

The rainfall data obtained from the WRF model is cumulated to daily timesteps and compared with the data from IMD, as shown in Fig. 6. Fig. 6 (a) shows the difference in daily rainfall at four locations (blue triangles in Fig. 1) between the WRF model and IMD data. It is observed that the rainfall difference is not substantial from 9 June 2013 to 14 June 2013 but increases significantly during the days of heavy rain on 15, 16, and 17 June 2013. A similar trend is also observed from the interpretation of Fig. 6 b. The hourly timeseries of rainfall from WRF compared with the GSMap data Fig. 7 a, b, and c shows an overestimation of WRF rainfall. However, the trend of the rainfall simulated by WRF agrees with the satellite data Fig. 7 b and c. Considering the coarse resolution of satellite data, the WRF model shall be used to discretely obtain rainfall intensity-duration thresholds of debris flows in data scarce-regions.

The estimated volume of debris flow based on the empirical equation by Chang et al. (2011) is 2320000 m$^3$. The debris flow volume as computed by the numerical simulation is 2820000 m$^3$ (see Fig. 10). This value is the closest to the empirical estimation achieved by the numerical model after performing rigorous parametric simulations considering a range of values for the parameters d50, $\delta_e$, and $\delta_d$. For validation, the debris flows extents obtained from the numerical analysis are superimposed over the debris flow polygons mapped by Martha et al. (2013) (see Fig. 10 a). We statistically evaluated the spatial accuracy of debris flow inundation, comparing it with the inventory. The model correctly classifies the debris flow areas and non debris flow areas about 63.3 percent. Whereas, the model incorrectly classifies the debris flow and non debris flow areas, about 36.7 percent. Most of these false positives are spread over hillslopes away from river channels. This could be due to the different debris flow mechanisms i.e., distinguishing hill slope originated debris flows vs runoff generated debris flows (Iverson et al., 1997; Kean et al., 2013; Tang et al., 2019) which the model did not simulate satisfactorily. The model's accuracy is 63 percent. The Cohen's Kappa of the prediction is 0.26, which is a fair agreement (McHugh, 2012). Please see more details of the accuracy assessment in the Supplementary.

The major devastation in Kedarnath occurred on 16 June 2013, as per sources (Champati Ray et al., 2016; Dobhal et al., 2013). The triggering of the largest debris flow that caused the outburst from the Chorabari glacial lake occurred on 16 June

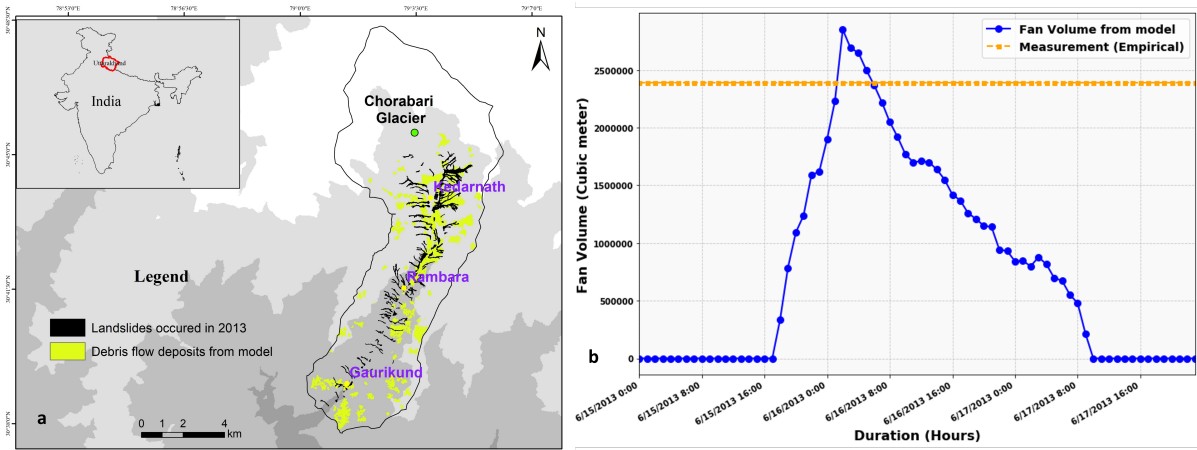

**Figure 10.** (a) Debris flows extents obtained from the numerical analysis are superimposed over the debris flow polygons mapped by Martha et al. (2013) and (b) Numerically estimated debris flow volume compared with empirical estimation.

2013 (Martha et al., 2015). Champati Ray et al. (2016) also identified that the major debris flow and reactivated landslides occurred on 16 June 2013. The numerical model results shown in Fig. 10 b show the largest volume of debris flow to occur on 16 June 2013 and then the sediment move further downstream. With these agreements in the volume and timing of debris flows from the model and actual scenario, the model shall be considered calibrated for the 2013 rainfall-induced debris flows.

From the parametric numerical simulations performed to identify the intensity-duration thresholds of debris flows in Kedarnath, it is found that debris flows occur under all eight rainfall intensities except for 10 mm/hr 15 mm/hr based on the results of numerical simulations. Fig. 12 shows the ID thresholds plotted in a 2D plane from all the eight numerical simulations. The difference in the arrival of debris flows under constant rainfall intensities is shown in Fig. 11. It is observed that the duration of rainfall needed to trigger the debris flow decreases as the intensity of rainfall increases because higher intensities of rainfall trigger debris flows quickly. For a given rainfall intensity (I) in mm/hr., the duration at which the debris flow arrives at the confluence of the river is considered as D (hours). Through this analysis, an ID threshold is obtained for the debris flow event using the material parameters similar to the calibration of the numerical model (see Fig. 12). The ID threshold shown in Fig. 12 is fitted using Eq.3 with $\alpha$ and $\beta$ being 33.96 and -0.47 respectively.

## 5 Discussion

### 5.1 Rainfall Intensity-Duration thresholds for landslides in the Himalayas

Berti et al. (2020) clarified the physical significance of the ID thresholds relationship for runoff-induced debris flows. In India, studies that determine the intensity and duration of triggering rainfall for landslides, particularly in the Himalayas (Dikshit et al., 2020; Teja et al., 2019; Kundalia et al., 2009; Kanungo and Sharma, 2014) are available. The use of daily, 3-day, and 15-day cumulative rainfall for threshold determination is a major similarity between these studies. The cumulative thresholds

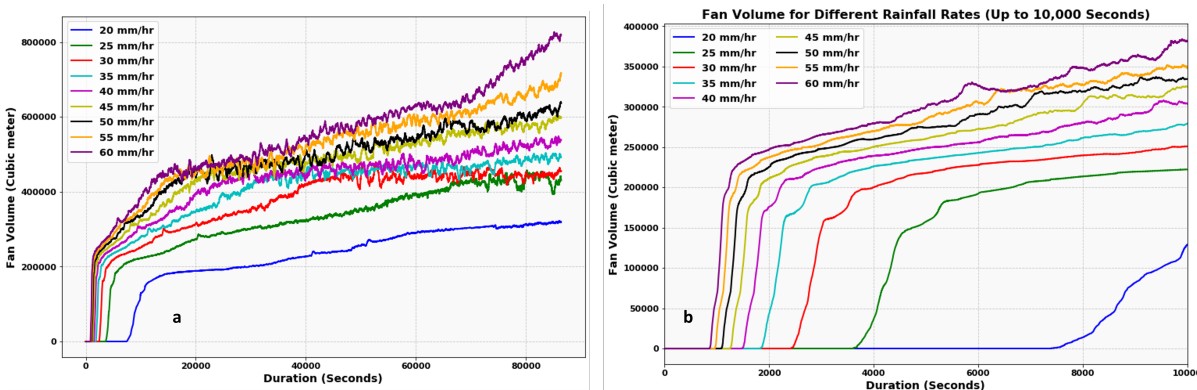

**Figure 11.** Initiation of debris flows in response to various constant rainfall intensities from 20 mm/hr. to 60 mm/hr.(a) Plot showing comparison of debris flow fan volumes generated by various rainfall intensities up to a duration of 86400 seconds = 24 hours (b) magnified view of Fig. 11 (a) up to 10000 seconds duration.

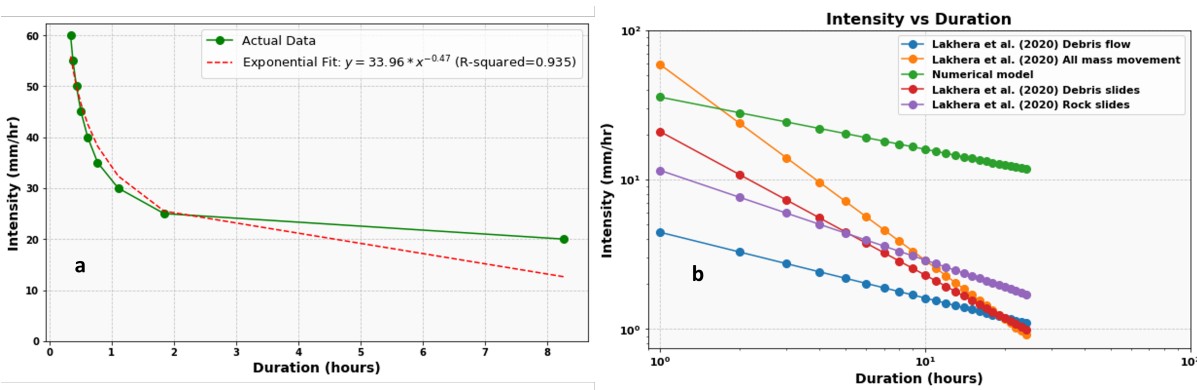

**Figure 12.** Numerical model derived intensity duration threshold for early warning of debris flows in Kedarnath catchment (a) ID curve compared with Lakhera et al. 2020 (b) ID curves and comparison plotted in log-log plot.

could help predict or forecast rainfall-induced shallow landslides. However, for debris flows, the thresholds must be determined using hourly rainfall data. This is due to the fact that runoff-induced erosion occurs during extreme rainfall events that last only a few hours, like in Kedarnath, though the rainfall was continuous for 3 days, the debris flow episodes lasted only between three to four hours (Champati Ray et al., 2016). Te-LEWS for debris flows in the United States, Italy, and Japan also use hourly
rainfall data to forecast the ID thresholds of landslides (Godt et al., 2006; Guzzetti et al., 2008; Mirus et al., 2018; Guzzetti et al., 2020; Piciullo et al., 2018; Baum and Godt, 2010; Osanai et al., 2010; Ju et al., 2020; Alfieri et al., 2012). These countries also use radar-based rainfall forecasts to help predict precipitation magnitudes six hours in advance. However, in India, where such radar-based forecasts are yet to be available, using WRF models is unavoidable and prudent.

In India, the authors have found studies that consider the sub-daily rainfall as a threshold (Lakhera, 2015; Lakhera et al., 2020). We preferably compare the ID thresholds obtained from the numerical model with the results of Lakhera et al. (2020) (see Fig 12(b). It is seen that, the thresholds estimated in this study is higher than all the thresholds estimated by Lakhera et al. (2020). The possible reason for the higher magnitude of thresholds could be due to the nature of the debris present in the Kedarnath catchment, and these thresholds correspond to only extreme rainfall events. Whereas the thresholds considered by previous studies incorporate landslides from different catchments. The divergence in ID thresholds between our numerical model and the study by previous work underscores the nuanced nature of rainfall-induced debris flows, influenced by regional variations. The higher thresholds identified in our study for the Kedarnath catchment might be indicative of its distinct characteristics, emphasizing the importance of tailoring threshold analyses to the specific geological and hydrological attributes of the study area.

## 5.2  Limitations of the study

This study uses a hydrological model to simulate debris flow dynamics from hourly rainfall. The model considers moisture content-dependent seepage through the unsaturated debris, runoff and overland flow based on Horton's equation, erosion based on critical thresholds, and debris flow deposition based on sediment concentrations. The following are the study's few limitations. The model uses of an open-source DEM with a resolution of 12.5 m. The effect of DEM resolution on debris flow routing is not considered. However, Boreggio et al. (2018) found that re-sampling the DEM to a finer resolution had no significant effect on the model results but this was not considered in this study. This study's erosion equation is a simplified representation of various erosion mechanisms occurring over loose material deposits in the hillslopes and in the channel (Iverson et al., 1997). The numerical model for debris flows that we used and developed in this study is a single-phase flow dynamic model compared to models that uses a two-phase flow assumption (Pudasaini, 2012; Bout et al., 2018). Through a digital elevation model, it simulates debris flow dynamics from rainfall, runoff, erosion, and deposition of debris flow. An infinite slope model underpins the stability theory. This study's numerical modelling strategy has several limitations due to the simplified numerical approaches and empirical equations. It is possible to introduce more robust erosion modules to simulate the processes in channel systems (Egashira et al., 2001; Berti and Simoni, 2005; Medina et al., 2008; Quan Luna et al., 2011). However, this needs extensive fieldwork and instrumentation of advanced monitoring systems which may be possible in future studies. In future works, we aim to establish an insitu monitoring system in Kedarnath valley to understand the controlling factors and dynamics of debris flows in the Himalayas. This study has explored the use of numerical modelling to derive the ID threshold for debris flow early warning using the 2013 Kedarnath disaster despite the above limitations.

## 6  Conclusions

Rainfall induced debris flows are catastrophic geohazards that multiply in number during intense rainfall events. Rainfall intensity-duration (ID) thresholds are used in early warning systems for shallow landslides and debris flows at the territo-

rial scale. In India, Te-LEWS have trouble predicting and correlating sub-daily time series of weather for landslides in the Himalayas because of instrumental and operational difficulties.

    In conclusion, this study aims to contribute a thoughtful framework to address the challenges faced by Te-LEWS in predicting rain-induced debris flows in the Himalayas. By integrating a spatio-temporal numerical model with the WRF model, we aim to offer a practical solution within existing operational constraints. Our approach involves careful model calibration using data

from the 2013 landslides in Uttarakhand, India. This calibration serves as a grounding mechanism that tries to align the model with the unique geological characteristics of the region. A subsequent validation process that compares our trigger intensity-duration (ID) thresholds with established findings from the literature modestly confirms the reliability of our model in a specific context. Using the WRF model at 1.8 km * 1.8 km resolution improves our ability to analyze hourly precipitation patterns and provides a finer understanding of the temporal dynamics. Our survey of constant rainfall intensities (ranging from 20 mm/hr

to 60 mm/hr) modestly confirms our established thresholds against earlier research findings for Uttarakhand. In an attempt to address the challenges faced by Te-LEWS, our integration of the WRF model into an early warning system based on ID rainfall thresholds has a potential for improved forecasts. While we acknowledge the limitations of our study, particularly with respect to the inherent uncertainties in weather forecasting and changing terrain conditions, we hope that our approach will make a modest contribution to the ongoing discussion on increasing the effectiveness of early warning systems for rainfall

induced debris flows. As we conclude, we recognize the ever-evolving nature of meteorological and numerical modeling and their application to landslide early warning. This study may pave the way for further refinements and applications of Te-LEWS in various geological settings.

*Code and data availability.* The spatio-temporal numerical model for debris flows developed and used in this study is publicly available together with the datasets for modelling and analysis. It can be accessed from our GitHub repository: https://github.com/srikrishnan-ss/

aschpeired.git.

*Author contributions.* SD performed the revised analysis, post-processed the numerical model outputs, prepared the plots and contributed to the revised design of this study. SSS designed the study and performed the debris flow numerical modelling and ID threshold analysis. SD and SSS contributed equally. PS performed the Weather Research and Forecast (WRF) model analysis. APY performed the debris flow volume calculation and contributed to the design of this study. TRM prepared the landslide inventory. SS provided the required precipitation

datasets. All the authors have contributed to the writing of this manuscript.

*Competing interests.* The authors declare no competing interests

*Acknowledgements.* The authors sincerely acknowledge the Indian Institute of Remote Sensing (IIRS), Indian Space Research Organisation (ISRO) for grant-in-aid support through a research project under the Disaster Management Support Program (ISRO-DMSP) Grant No. IIRS/DO/DMSP-ASCB/AS/2022/01 and IIR-1902-DMC/22-23. The authors thank Mr. Akshat Vashistha and Mr. Manish Dewrari, Centre of
305 Excellence in Disaster Mitigation and Management (CoEDMM), IIT Roorkee for their assistance during fieldwork in Kedarnath. The authors acknowledge the help of Mr. Rajendra Singh Rawat and Mr. Anandu P, CoEDMM IIT Roorkee in post-processing the WRF model outputs. Ms. Anamika Sekar and Ms. Shivani Joshi, CoEDMM IIT Roorkee are thanked for help in processing some figures. The corresponding author sincerely acknowledges Late Prof. Theo van Asch for his time and insightful discussions during the course of development and programming of the debris flow numerical model. All authors pay tribute to Late Prof. Theo van Asch for his contributions in research on landslides and
310 debris flows in particular. The authors sincerely thank the reviewers of this manuscript for providing critical and constructive comments which helped to improve the quality.

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
