# Peer review of "Numerical model derived intensity-duration thresholds for early warning of rainfall-induced debris flows in a Himalayan catchment"

_Natural Hazards and Earth System Sciences, 2022_

## Author Response (AR1)

**NHESS**

**Ref: NHESS-2022-297**

**Title: Numerical model derived intensity-duration thresholds for early warning of rainfall-induced debris flows in a Himalayan catchment**

**Response to Referee #1**

| Ref. | Comment | Reply |
|---|---|---|
| 1 | I read your manuscript and I appreciated the idea and motivation, but I was left with many questions. | Thank you for your careful consideration and thorough evaluation of our manuscript. We sincerely appreciate your encouragement towards the idea and motivation of this article. Please see the detailed responses below to each of your comments. |
| 2 | I do not have a problem with the use of WRF model data to predict hourly rainfall but the prediction needs to be validated with some field data. | Thank you for your commendable suggestions. We agree with your concern here. We validated the prediction of rainfall using gauged data and the prediction of debris flow volume using empirical data based on field work. In our revised manuscript you will find these major revisions. |
| | I understand no hourly data are available in the catchment under study, but are there any data from adjacent areas to verify a degree of correlation? | Thanks, we do not have any ground-based rainfall measurement in hourly timestep to validate the WRF outputs. However, considering your suggestion here, we validate the cumulative daily rainfall of WRF outputs with available ground-based precipitation datasets from the India Meteorological Department (IMD). The validation results in > 80% accuracy both spatially and temporally. Kindly see the revised manuscript section "rainfall data validation". |
| | This verification should be done not only for the specific event but in general (e.g., over a whole year) to prove that your approach can be extended and used as a prediction tool. | In addition, we validate hourly rainfall from the WRF model with spatially and temporally satellite-derived precipitation data. We also agree to extend the verification over a year (possibly). However, due to computational resource limitations we do not perform a over a whole year simulation of WRF model at this moment. |
| | Consider, for example, if your WRF predictions are systematically an overestimation. You still would capture the debris flow events you were seeking, but you would also launch many false alarms. | We agree to the point that WRF predictions may be systematically an overestimation. Of course, this will launch many false alarms of debris flows. However, since using WRF models for landslide early warning is new to the study area, a conservative approach will help first. We would have to tune the model to estimate the rainfall forecast properly. |
| 3 | Further, I understand you validated the approach using a rainfall event during | Thank you for this very comprehensive and helpful comment. |

| | | which a disaster was actually triggered. This is ok but it is only half of the validation, namely a true positive identification in space and time. | |
|---|---|---|---|
| | | What about another event with similar characteristics that did not trigger debris flows in that catchment? Or the same event but in an adjacent catchment where no debris flows occurred? To be usable as a warning system, your approach should also be able to identify true negatives in space and time. | During the 2013 North India Floods, more than 6000 landslides were triggered in Uttarakhand, India. Though it is difficult to identify a catchment where no debris flows occurred due to this extreme rainfall event, we try our best and identified a catchment with fewer debris flows and performed the simulation.

The model also predicts no/a smaller number of debris flows for the given region. Thanks to the reviewer, this has also given us confidence in the model.
However, we have not included this in the revised manuscript but instead have shown it in supplementary.
Please see the details on the supplementary page no 6 -7. |
| 4 | | Further, you had to make assumptions due to lack of field data (e.g., on the pre-rainfall moisture) but you did not discuss how reasonable your choice was or how sensitive the result is to a change in the chosen value. | Thank you for your careful examination and constructive suggestions. We agree with your concern here.

You are correct in pointing out that we had to make assumptions due to a lack of field data. Since the initial conditions could be sensitive to the triggering time of debris flows, we had to decide that carefully.
We run a decadal simulation of rainfall-runoff-infiltration using daily timesteps of rainfall (data from IMD) from 2003 to 2013. We used the initial moisture content from the results.
Please see supplementary page 7 for details on the initial moisture simulation. |
| | | In other words, where does the 5% moisture come from? Is it supported by field or lab experiments? What changes if you use a moisture of 0% or 20%? | Thanks for the thoughts. We also perform additional numerical simulations to test the sensitivity of initial moisture content and include these results in the revised manuscript.

However, we found that the initial moisture content sensitivity is less for the 2023 North India Flood event possibly due to the higher intensity and longer duration of rainfall. |
| 5 | | Finally, the DEM resolution. 30 m does not really seem great at your scale, with a catchment of few km. I agree that resampling cannot improve the result (because a smooth DEM remains smooth after resampling), but what about an actual high-resolution DEM that more closely follows the roughness of the morphology? If not available in this catchment, couldn't the authors study this sensitivity in another location with better data, to assure the reader that the result remains reasonable? | Thank you for this very useful suggestion. We run a parameter sensitivity analysis using different resolutions of DEM to assure the reader that the result remains reasonable.
In the revised manuscript, we have used a 12.5m resolution DEM. |

**Response to Referee #2**

| Ref. | Comment | Reply |
|------|---------|-------|
| 1 | **General comments** | Thank you for your careful consideration and very detailed evaluation of our manuscript. |
|  | Thank you for the opportunity to provide a peer review for this manuscript titled "Numerical model derived intensity-duration thresholds for early warning of rainfall-induced debris flows in the Himalayas" (nhess-2022-297). This study uses the Weather Research and Forecasting (WRF) model to estimate hourly rainfall time series at four meteorological station locations near the Kedarnath catchment, Uttarakhand, India, which record daily rainfall totals, during a debris flow event that occurred in June, 2013. A previous study mapped 120 debris flows resulting from this event in the catchment. This study estimates the volume of debris flows during the 2013 event from this inventory with an empirical relationship originally developed for Taiwan. The authors then use this estimated volume, along with an averaged precipitation time series across the four stations, to calibrate a numerical debris flow initiation and runout model for this event. With the calibrated model, the authors simulated cumulative debris flow volume with time as a function of average rainfall intensity for a range of intensity scenarios. The authors plot the time to initiate a debris flow against the average rainfall intensity in these scenarios to define an I-D threshold for the Kedarnath catchment. | We sincerely appreciate your time to provide constructive as well as critical comments on the manuscript, analysis, technical aspects and writing of this research article. We thank your encouragement towards the intriguing idea of this manuscript and identifying it within the scientific scope of NHESS. The authors agree this study need serious and substantial modifications in the analysis and manuscript to meet the standards for publication in NHESS and have performed the revisions thoroughly. Please see our detailed responses below to each of your comments sectionized in the order. |
|  | While the idea to use a weather forecasting model coupled with a numerical debris flow model for landslide early warning in regions without available hourly rainfall data is intriguing and within the scientific scope of NHESS, this study will need serious and substantial modifications to both the analysis and manuscript before it can be considered for publication in NHESS or any other journal. At this stage, this manuscript does not meet the standards for publication in NHESS. | |
|  | I summarize my main comments on the manuscript and the analysis here, and then provide more specific comments in the next sections. | |

Comments on Manuscript:

| Ref. | Comment | Reply |
|------|---------|-------|
| 1 | The abstract makes various statements that are not supported by references or analysis in the main text and does not report the key results. The abstract suggests that the estimated I-D threshold will be used in a LEWS, but this is not validated or sufficiently discussed in the main text. | Thank you for pointing the lack in the abstract writing. We add more citation wherever necessary and make the sentences relevant to the key results reported in the manuscript.

Following your suggestion, we discuss in detail the usage of I-D threshold method in a LEWS in the main text of the revised manuscript. |
| 2 | The introduction makes numerous incorrect statements, lacks sufficient supporting literature, and does not clearly define a research question or objective. | Thanks, we carefully examine for any incorrect statements in the introduction. Thank you for pointing out the lack of clarity in the research question/objective. We address this in the revised manuscript. |
| 3 | The methods section does not provide sufficient information to reproduce the analysis or to evaluate the validity of the results. It does not meet basic quality standards, such as defining all parameters. Some key parameters for the debris flow model are reportedly set by "calibration and back analysis," but the details of said calibration are missing altogether. | Thank you for this useful suggestion. We provide detailed information about all the analysis performed.
Details of the calibration and back analysis are included in the revised version. |
| 4 | The results section, which is only 12 lines long, includes methods descriptions and does not describe the key results of the study | Thank you again. We extend the results section detailing every aspect of the research describing the key results and move the method description to the methodology section. |
| 5 | The discussion section does not discuss the results of the study or their implications. While it points to some limitations of the analysis, it importantly fails to evaluate the usefulness of the identified threshold for early warning, as suggested in the title. | Thank you very much for pointing this flaw. We improve the discussion section to evaluate the usefulness of the identified threshold for early warning. |
| 6 | The conclusions section repeats introductory and methods material but does not reach substantial conclusions based on the study's results. | Thanks, we agree and revise the conclusion section based on the results obtained from this study. |

Comments on Analysis:

| Ref. | Comment | Reply |
|------|---------|-------|
| 1 | This study uses the WRF model to estimate hourly rainfall across the catchment during the 2013 debris flow event at 1.8 km resolution. However, although daily meteorological station observations are available at four locations near the catchment, there is no validation or analysis of how well the simulated hourly precipitation totals match the daily totals at each station. Such a validation is required, particularly because this study proposes using the | Thank you for pointing it out. We totally agree. We do not have any ground-based rainfall measurement in hourly timestep to validate the WRF outputs. However, considering your suggestion here, we validate the cumulative daily rainfall of WRF outputs with available ground-based precipitation datasets from the India Meteorological Department (IMD). |

| | | | |
|---|---|---|---|
| | WRF model as an approach for areas without hourly data. | | |
| 2 | Despite running a spatially explicit weather model and a spatially explicit debris flow model, the authors have chosen to drive the debris flow model of the 2013 event using an averaged hourly precipitation time series at four stations with an elevation difference of ~5000 m. I strongly question this choice, as such an elevation difference likely leads to substantial variations in rainfall intensity across the catchment (Destro et al., 2017; Iadanza et al., 2016). I recommend taking advantage of the available spatially explicit WRF rainfall estimates to drive the debris flow model. | Thank you for your useful critique. We also agree and following your recommendation, in the revision we use spatially explicit rainfall timeseries (from WRF) for the time period to drive the debris flow model. | |
| 3 | The debris flow model was calibrated using an empirical estimate for debris flow volume during the 2013 event. The baseline volume estimate was made using an empirical equation originally developed for Taiwan, which is of questionable validity in this different setting. Potentially resulting from this or other sparsely documented modeling choices, the debris flow model substantially overpredicts debris flow areas compared to the mapped inventory, but this is not discussed. Meanwhile, an analysis of how well the model could predict debris flow timing during the 2013 event is missing. Such an analysis is crucial for evaluating this model's usefulness for early warning. | Thanks, we agree to both these critiques. First, we validate the area-volume estimation using different empirical equations other than the one used in the first version of the manuscript.

We also estimate the accuracy of the debris flow model outputs using True Skill Statistics and Chi-square tests.

Regarding the timing of debris flow triggering, we cross check the initiation time from the model with the one reported in the literature. However, different debris flows out of the total 120 might have been triggered in different time for which we do not have the data and the model is not able to simulate them differently. We discuss these in the revised manuscript. | |
| 4 | The scenario analysis conducted to determine points for the I-D threshold relies on constant precipitation intensities, which is unrealistic for any rainstorm. The shape of the hyetograph is important for determining whether or not landslides are triggered (D'Odorico et al., 2005), and I therefore question whether a constant precipitation intensity can sufficiently represent triggering rainfall for estimating warning thresholds. A potential alternative approach could be to use a precipitation generator to run a suite of scenarios and use these to investigate triggering thresholds (Thomas et al., 2018). | Thank you very much for pointing out a lack in our understanding here. To check the shape of the hyetograph, we compare the WRF outputs with satellite derived hourly precipitation.

We are not familiar with the use of precipitation generator. However, we will explore the option suggested here. | |
| 5 | No uncertainty of the identified I-D threshold is estimated or discussed. | Thanks, yes. We perform an uncertainty analysis considering the over or underestimation of rainfall intensity and duration. | |
| 6 | Although this threshold is apparently intended for use in a territorial LEWS, | Thank you for your right comment. As the LEWS is in a prototype stage, we are still not in a | |

| | there is no validation of this threshold's performance for early warning. | position to evaluate the threshold's performance for early warning. |

Specific Comments:

| Ref. | Comment | Reply |
|---|---|---|
| 1 | **Title**

The title is clear, but promises early warning applications, which are not analyzed and barely discussed in the text. Himalayas suggests a broad region, please specify (e.g. "a Himalayan catchment"). | Thank you. We agree and modify the title as "***Numerical model derived intensity-duration thresholds for early warning of rainfall-induced debris flows in a Himalayan catchment .***" |
| 2 | **Abstract**

• Line 2: Many early warning systems at the territorial scale do not use I-D thresholds (Guzzetti et al., 2020; Scheevel et al., 2017; Peruccacci et al., 2017).

• Line 3-4: Introduction does not provide evidence for this claim.

• Line 5: Specify what the numerical model does. Does it only apply to extreme rainfall? If so, how do you define "extreme"? Not supported in the text.

• Line 7: Which input boundary condition? This is not described in the methods.

• Line 8: Specify which model.

• Line 9: Glossary not mentioned in methods. | Thanks for the suggestion. Yes, we amend the sentence providing details of LEWS do not use the I-D thresholds.

Please see page 1 lines 2 – 3 in the revised manuscript.

We amend the introduction with evidence to this claim.

Please see page 2 lines 41 – 43 in the revised manuscript.

Thanks for the suggestion. The numerical model simulates erosion/debris flow triggering for any given rainfall intensity and not just for extreme rainfall. We include the definition of extreme rainfall based on IMD glossary. Relevant details are included in the revised manuscript and in the supplementary document.

Please see page 1 lines 9 – 10 in the revised manuscript.

Thanks. We include the details of the boundary conditions in the methods clearly.

Please see page 1 lines 9 – 10 in the revised manuscript.

Thanks, we specify the model in the amended version of the manuscript.

Please see page 1 line 9 in the revised manuscript.

The glossary is explained the supplementary in the revised version of the manuscript.

Please see page 5-6 in the revised supplementary. |

| | | • Line 11: Use of this threshold in a LEWS is not evaluated or sufficiently discussed in the main text. | Thank you. We briefly explain the LEWS which uses the I-D threshold. However, as the LEWS is in a prototype stage, we are still not in a position to evaluate the threshold's performance for early warning. |
|---|---|---|---|
| 3 | **Introduction** | | |
| | | • Line 14: Although the frequency and magnitude of extreme rainfall may be increasing, there is to my knowledge so far no empirical evidence that shows that disastrous debris flows have become more frequent. These citations do not show it. Please adjust wording or include the relevant literature. | Thank you for your critical suggestions in the content, writing and presentation of the Introduction. Thanks, we agree to your opinion. We rephrase the wordings and include relevant literature to possible support the inference. Please see page 1 line 18 in the revised manuscript. |
| | | • Line 17: Debris flow *impacts*. Non-structural measures do not mitigate debris flows. | Thank you. Yes, non-structural measures do not mitigate debris flows but may help in reducing the impacts caused by debris flows. We rephrase it accordingly. Please see page 1 line 21 in the revised manuscript. |
| | | • Line 19: Adapt to what? Please specify. | Adapt to practices for efficient early warning. Please see page 2 line 22 in the revised manuscript. |
| | | • Line 20: These cover some regions, but few cover entire nations. Please reword. | Thanks, we reword as suggested. Please see page 2 lines 24-25 in the revised manuscript. |
| | | • Line 21: This statement is incorrect and needs citations. I-D thresholds are rarely estimated using forecasts, but are usually determined using observed rainfall. | Thank you, we agree and correct it accordingly. Please see page 2 line 26 in the revised manuscript. |
| | | • Line 24: Needs citations. Consider (Intrieri et al., 2013; Segoni et al., 2018; Stähli et al., 2015) and references therein. | Thanks for suggesting, we include these citations in the revised manuscript. Please see page 2 lines 28-29 in the revised manuscript. |
| | | • Line 30. This statement appears to be incorrect. Figure 6 of (Mathew et al., 2014) presents an I-D threshold with points with <24 hour durations. | Thanks, the thresholds used in their study uses 3 hourly rainfalls from TRMM 3B42 V.6 rainfall data. However, the actual LEWS operated by National Remote Sensing Centre (NRSC), Indian Space Research Organisation (ISRO) uses daily as well as multiple days antecedent rainfall |

| | | | based on Mathew et al., (2014). We rephrase as suggested. |
|---|---|---|---|
| | | | Please see page 2 lines 36-38 in the revised manuscript. |
| | • Line 32. This statement needs references.
• Line 34. References. | | Link to NRSC-ISRO LEWS. |
| | | | Thanks, we include references in Line 32 and Line 42. |
| | | | Please see page 2 lines 41-42 in the revised manuscript. |
| 4 | **Study area and characteristics of the disaster**

Line 58: What is a fragile landscape? Perhaps prone to slope failures? | | Thanks, yes, by fragile landscape we mean the weakened geological formations and lithology susceptible to slope failures. |
| | | | We revised the sentences. Please see page 3 line 66. |
| | • Line 59-60: Show these faults on Figure 2. | | Thanks for the suggestion. We include the faults in Figure 2. |
| | | | Please see the revised Figure 2 in page 5. |
| | • Line 61. The major rock types listed here do not match Figure 2. Please revise.
• Line 63. Please define "extreme rainfall" in this case. This suggests that over 6000 landslides occurred, but many fewer are shown in Figures 1 and 2. Why? | | Thanks for the observation, we revise the legend in Figure 2. |
| | | | Thanks, we include the glossary of India Meteorological Department (IMD) definition of extreme rainfall in the revised manuscript. 6000 landslides occurred al over Uttarakhand but only 120 occurred within the study area. We include a map of Uttarakhand showing all 6000 landslides. |
| | | | We revised the sentences. Please see page 3 line 73 |
| | | | Please see revised Figure 2 in page 5. |
| | • Line 65. Reference for number of casualties and economic impacts needed. | | Thanks, we include these references in the revised manuscript. |
| | • **Figure 2.** It is difficult to distinguish the red and black debris flows / slides in 2b. | | Thanks for the suggestion. We revise the symbology in Figure 2. |
| | • **Figure 3.** Please label Chorabari glacier lake on Figures 1 and 2. | | Thanks to your suggestion, we include the Chorabari glacier lake in Figure 1 and 2. |
| 5 | **Data and methods**

• Data and methods general comment: this section does not provide sufficient detail to reproduce or evaluate the results, and is somewhat difficult to follow. Particularly, | | Thanks for your careful examination and suggestion for improvements. We include detailed information regarding the model parameter or reproduction of the results. Defining all parameters in the text, cite the models |

| | | not all parameters are defined in the text, models are mis-cited, datasets are not cited, and key modeling choices and approaches are not described. This section must be more thorough. | properly, providing citations to the datasets, we try to make this section more thorough. |
| | | • Line 75. Please explain briefly what this model is, what it does, and what it is used for in this study. Model needs a citation. | Thanks to your suggestion, we provide brief description of the model with citations in the revised manuscript.

Please see page 3-5 lines 85-90 in the revised manuscript. |
| | | • Line 77. Figure reference wrong, please double check and correct throughout the manuscript. | Thanks for pointing out this mistake. We carefully check the Figure references throughout the manuscript. |
| | | • Line 79. I infer that Locations 1-4 are meteorological stations that record daily rainfall, but this needs to be specified in the text. | Thanks for the suggestion. Included the locations from where we infer meteorological data in the text. But these are not meteorological stations. Explanations are given in the revised manuscripts.

Please see page 4 lines 93-96 in the revised manuscript. |
| | | • **Figure 5.** I am not an expert in weather models, but I suppose that the information presented in this figure would not be sufficient to reproduce the results. I recommend creating supplementary tables that specify the inputs used for all models. All datasets require citations. | Thanks for the valuable suggestion. We include a very detailed information to reproduce the WRF model analysis and include primary information in the main manuscript and secondary data in the supplementary. We also cite the datasets retrieved from secondary sources properly.

Please see page 5 lines 99-101 in the revised manuscript. |
| | | • **Figure 6.** From this figure, I would like to be able to evaluate whether the simulated hourly rainfall time series at each of the stations matches the daily records. Please rescale Fig 6a such that this is possible, or better yet, perform such a validation. | Thanks for the useful suggestion. We reformat Figure 6..

Please see page 13 lines 194-198 in the revised manuscript. |
| | | • Line 80. Does the WRF model only output one possible time series? Or did you somehow select this time series from a range of options? How sensitive are these results to inputs and modeling choices? | Thanks for the question. For rainfall the WRF model provide one time series per pixel/resolution of the model. These results are sensitive to the initial boundary condition as well the opted physics to run the model. We document these modelling choices clearly. |

| | | Please document any modeling choices or selections. | Please see Figure 5 on page 8. |
| | | • Line 81. This states that the authors have averaged the hourly precipitation time series at the four station locations and used this to drive the debris flow model. I do not understand this choice. From Figure 1, I infer that between Locations 1-2 and 3-4, there are 5000 m of elevation difference. I would expect this to introduce substantial variability in rainfall intensities (Destro et al., 2017; Iadanza et al., 2016) and therefore do not expect an average to appropriately capture this event. I do not understand why, when a 1.8 km resolution time series over the catchment is available, this information was not used to drive the debris flow model. I would recommend taking advantage of this available information, but at the very least, a sound justification of averaging is needed. | Thank you very much for this valuable comment and suggestion. We agree with you. We opted to average out the rainfall at these four locations in order to reduce any uncertainties (both over estimations and under estimations). However, we understand from your suggestion that this may not be a good option considering the elevation difference.

We rerun all the debris flow modelling using spatially different maps for every one-hour timesteps to drive the debris flows.

To achieve this we had to restructure the coding of the model entirely allowing the input of spatially distributed map of hourly rainfall into the debris flow model.

We could see some improvements in the analysis outputs. Thank you very much for your recommendation.

The detailed spatial outputs of the WRF model is shown in Supplementary information.

Please see Figure S1 on page 1 in the supplementary. |
| | | • Line 84. Here, please also briefly describe what this model is, what it does, and what it is used for in this study. (e.g. "We use a numerical debris flow initiation and runout model…"). Siva Subramian et al., 2021 is a pre-print; this is not a sufficient citation. More detail is needed in this manuscript describing this model. | Thank you for this suggestion. We include all details of the model's governing equations with proper justification and citations. We add more detail about the numerical model in this manuscript instead of simply citing the previous literature.

Please see lines 123 to 137 in the revised manuscript. |
| | | • Line 91. Depth of soil or regolith is a very important parameter. How was this determined? Although Figure 8c plots "Soil Depth," this is not described anywhere in the text. The field work photos from Figure 3 do not suggest much soil development on these slopes, so is this actually regolith depth? | Thank you very much for your careful observation. We use Hengl et al. 2017 SoilGrids250m dataset to derive the soil depth or regolith thickness.

You are correct. The depth used is regolith depth.

We mention this in the revised manuscript.

Please see page 6 line 116. |

| | |
|---|---|
| • Line 94. "based in part" – which part? Again, the modeling strategy needs a more thorough description in this text. | Thanks again. We include a thorough description of the modelling strategy in the revised manuscript. |
| • Line 96. All parameter symbols in Table 1 need to be defined. | Thanks to your suggestion, we define all parameters defined in Table 1.

Please see lines 145-147 in page 9. |
| • Line 97. Why do you choose 0.05 m3/m3 as an initial moisture content across the entire catchment? Is this reasonable? From Figure 6, it appears that it had been raining in the days prior to the event, so dry hillslopes may not be an appropriate assumption. Why not spin up the hydrological model with time series from before the event, as this is likely available from the WRF model? | Thank you for your valuable suggestion. Since the initial conditions could be sensitive to the triggering time of debris flows, we had to decide that carefully.

We run a decadal simulation of rainfall-runoff/infiltration using daily timesteps of rainfall (data from IMD) from 2003 to 2013. We used the initial moisture content from the results. |
| • Line 99. How is the hourly rainfall data used with a time step in seconds? Please specify. | Thanks for this very important question. We explain the process of converting the input boundary condition into seconds from hours. However, in the revised manuscript we have changed the input boundary condition to hours to use the spatially explicit WRF. This process has resulted in improved accuracy of the results as well as sufficient decrease in time to run the model, thanks to the suggestion of reviewer. |
| • Line 102. What stream ordering system is referred to here? It would help to label these on Figure 1 or create another figure. | Thank you again, the catchment contains stream order 1, 2 and 3 and occasionally 0 order streams. We provide them in the caption of Figure 1 (revision). |
| • **Table 1.** Please describe all symbols used in the text. The "calibration and back analysis" for d50, delta_e, and delta_d is not described anywhere. This is a major issue, as the values of these parameters may have a strong impact on the results. Are these values justified considering your experience in the field? Judging from the photos in Figure 3, I'm not convinced that a d50 of 2.0 mm is appropriate, for example. In any case, some | Thanks for this another important question. Yes, you are correct and we agree with you. Through our field work, we observed a diverge range of grain sizes and quantified them using sample collection.

We also perform a sensitivity analysis and found that the revised d50 from the calibration is 46.9 mm as also mentioned in Table 1.

Please see Table 1 on page 9. |

| | | sensitivity analysis should be reported and discussed. | |
|---|---|---|---|
| | | • Line 104. Please use spellcheck throughout. See comments on I-D thresholds for LEWS in intro. | Thanks for the suggestion. We will perform spellcheck throughout the article when we finish our revised manuscript. |
| | | • Line 108. 'Berti…' - this should be moved to the discussion. | Thanks, we move this citation and corresponding discussion to the discussion. |
| | | • Line 110. The choice of "inter-event-time" varies widely between studies. Jiang et al., 2021 will have made one choice, but there are many others in the literature (Segoni et al., 2018). Please describe and justify your choice here. | Thank you for your comment here. We understand and agree IET could be different through diverse choices. However, we have not considered this in this study. |
| | | • Line 112. Please just describe your modeling approach here. The relationship between physical processes and statistical thresholds is material for the discussion. | Thank you very much for your valuable suggestion. We describe the relationship between physical processes and statistical thresholds in the discussion elaborately. |
| | | • Line 116. There is no methodological description of how the model was calibrated "above". This must be added. | Thanks. We include a methodological description of the model calibration. |
| | | • Line 117. I would make it very clear that you are now moving away from the 2013 event and into scenario analysis. This was hard for me to follow. | Thanks for your comment. Once the model is calibrated for the event, we run simulations using constant rainfall intensities to derive the I-D thresholds. Apologies for the confusion here, we provide detailed information in the revised manuscript.

Please see lines 216-220 on page 14. |
| | | • Line 119. Please specify what confluence is referred to here. | The confluence referred to herein is the Gauri Kund, shown in Figure 1 and Figure 2. |
| | | • Line 119-120. This method needs much more explanation. There are many statistical methods in use to establish I-D thresholds (Segoni et al., 2018, 2014; Staley et al., 2013; Brunetti et al., 2010; Scheevel et al., 2017). How do you select the threshold here? | Thanks for the very important question. The statistical methods work only when we have an abundant data of rainfall intensities and debris flow occurrences. Our method actually supplements the statistical analysis by providing triggering intensity of debris flows so that any further approach shall be used to determine the threshold. We explain our choice in the revision.

Thank you for this useful suggestion. For clarity, we restructure the scenario analysis from the 2013 event. |

| | | |
|---|---|---|
| | • Line 120. I was confused at this point that the text moves back to the 2013 event. I would recommend restructuring to separate the analysis of the 2013 event from the scenario analysis for the I-D threshold.

• Line 126. What values were used for I, D, and C_r in this equation? How did you define the rainfall event? I do not necessarily expect an empirical equation originally developed for Taiwan to be a reasonable approximation of debris flow volume in the Himalaya. Is such an equation transferable? Why? This needs to be discussed.

• **Figure 9.** It is not clear if this is a schematic figure or if it is results. If it's a schematic, please note that there is rarely such a clean separation of non-landslides and landslides, such that there are often rainfall events that exceed the threshold but do not trigger landslides.

• Line 128. Geological Index based on lithology needs explanation and documentation of values used. | Thank you very much. Your point is valid and true. Since we lack the true volume of landslides in this study area, we had to rely on these empirical estimates. First, we validate the area-volume estimation used in the first version of the manuscript.

We discuss them in detail in the revised version.

Please see lines 181-192 on page 13.

Thank, we agree. This is a schematic figure and we agree to your point that clean separation of non-landslides and landslides is not possible. We amend this figure in the revised manuscript using established ID thresholds.

We include the detailed method of selection of GI based on lithology in the revised manuscript.

Please see lines 182 on page 13. |
| 6 | **Results**

• General comment on the results section: This section is much too short, and does not describe the key results of the study. These should be presented for the reader.

• Lines 130-136. As I mentioned previously, this calibration needs to be documented in the methods section. The similar volume estimates are by design, as the numerical model was tuned to achieve this. However, judging from Figure 10, the model substantially overestimates the spatial area of debris flow deposits. I question whether such a model can be | Thank you for your careful comments on the results. Following your suggestion, we revised the result section describing all the key results of this study.

Thank you very much. We totally agree with your point here. Similar volume estimates are by design, as the numerical model was tuned to achieve this.

Overestimation is an issue in terms of spatial extents. We include the True Skill Statistics bases test to test the accuracy of the model. We also discuss the reasons and implications of the overestimation of spatial extents of the model in the revised version of the manuscript. |

| | | "considered calibrated." At the very least, this overestimation must be discussed. I would also like to see evidence that the numerical model can sufficiently reproduce debris flow *timing* during the 2013 event, not just volume. This is key if such a model is to be used for warning. | Thank you for your comment on timing of debris flows. We do agree this is very significant in terms of early warning. We compare the timing of debris flow initiation with the recorded timing of actual events from the literature. We include these details in the revised manuscript. |
| | | • Line 137. The 10 mm/hr plot should be shown in Figure 11. I am not convinced of the choice of intensities. First, using a constant intensity over the course of the scenario is unrealistic, even if I-D thresholds are often based on average intensities. As we can observe from Figure 10, intensities over the course of the 2013 event varied, and the average intensity was certainly less than 20 mm/hr, perhaps less than 15 mm/hr. The peak intensity at any location during this event was less than 40 mm/hr (Figure 6), but the scenarios continue up to 90 mm/hr. Since the shape of the hyetograph influences landsliding (D'Odorico et al., 2005), I question whether the choice of a constant intensity can sufficiently capture triggering rainfall here for use in a warning threshold. An alternative approach could be to use a rainfall generator to run many scenarios and use those to investigate thresholds. See, for example, (Thomas et al., 2018). | Thank you for your critique on the choice of constant rainfall intensities. The thresholds for a study area should be accountable for any given rainfall events and not just one event i.e., 2013. That is the reason we use constant rainfall intensities from 10 mm/hr. to 90 mm/hr. We agree to your point partially however using a rainfall generator as suggested in the reference Thomas et al., (2018) is beyond the scope of this study but we will try to consider this in future works. |
| | | • Line 143. It is not clear what event is referred to here, is it the 2013 event, or is this threshold valid for any rainfall event. "Material parameters similar…" were these adjusted after calibrating the model or are they the same? | Thanks for your point to bring out more clarity. The resultant threshold should be valid for any rainfall event. We include a detailed explanation to clarify the reader. |
| | **Discussion** | | |

| | |
|---|---|
| • Lines 148 – 150. These require references. | Thanks, we include relevant references. |
| • Line 149. This argument states that previous thresholds are insufficient, but this study has provided no evidence that the estimated threshold would perform better in a warning system. Such evidence is required to support this argument. | Thank you, we rephrase the arguments more politely addressing the limitation of previous thresholds and provide evidence of the improvements provided in this study. |
| • Line 151. This argument states that runoff induced erosion occurs during extreme rainfall lasting only a few hours, but the 2013 event studied here appears to have lasted for days. This is a break in logic. | Thank you for pointing out logical mistake. We rephrase the sentence supporting our argument. Please see page 15 line 236 – 238. |
| • Line 152. See comments in intro on LEWS in other countries. Also, ID thresholds are estimated, not forecasted. | Thanks, rephrased as suggested. |
| • **Figure 12.** The comparison between the Lakhera et al., 2020 threshold and the threshold estimated in this study is not valid. The Lakhera et al., 2020 threshold plotted here is specified as $I_{max}$, whereas the threshold estimated in this study is based on average intensity. Lakhera et al., 2020, specify thresholds for debris flows and debris slides, but the threshold plotted here is for all mass movements. | Thank you for pointing out the invalidity of comparison. In the revision, we include a detailed comparison of the thresholds derived from this study with exiting thresholds available from the literature including Lakhera et al., 2020. |
| • Line 154. In the introduction, Mathew et al., 2014 is cited, which provides an I-D threshold that appears to be based on hourly data. This would be an additional point of comparison. Furthermore, as this is a publication for an international journal, a comparison of these results with the international literature is warranted. (Guzzetti et al., 2008; Segoni et al., 2018) are starting points. Importantly, there should be a discussion of why the results found here may be similar or different to other results reported in the literature. (Bogaard and Greco, 2018) may be helpful for this. | Thank you again for this useful comment and suggestion. We agree and compare the threshold with Mathews et al. (2019) and other international literature as suggested. However, these are for different terrain with various geology, so we plot the main comparison of our result compared to Lakhera et al. (2020). |

| | | |
|---|---|---|
| | • Overall comment on the discussion: Since this study intended to estimate I-D thresholds for early warning, there must be a discussion of performance for early warning, but this is missing altogether. At a bare minimum, would this threshold have successfully warned for the 2013 event? How often would the threshold be exceeded otherwise, resulting in false alarms? Is there any case in which missed alarms would occur? (Staley et al., 2013) could be a starting point for considering this. | Thank you very much. We try to include a discussion of performance of early warning for the 2023 event. However, we are not in a stage of investigating missed alarms. Anyhow, we will explore the option suggested here following Staley et al., (2013). |
| | • Additional overall comment on the discussion: many limitations are listed, but without discussing how these might impact the identified threshold. Indeed, many modeling choices were made throughout the study, and these may induce uncertainty in the threshold, but that uncertainty is not quantified or discussed. The discussion should address these sources of uncertainty. | Thank you very much. We include detailed discussion on the limitations and their impact would be on the identified threshold quantifying the uncertainty in each step. Also, we address the source of uncertainties and the possible ways to address them. |
| | **Conclusions**

• Overall comment on the conclusions: The conclusions section repeats introductory and methods material, but does not reach conclusions based on the results presented in this study. The final statement that the approach presented in this study is promising for establishing Te-LEWS in new geological settings is not supported by the analysis or results presented in the study. | Thanks again. We elaborated the conclusion part detailing the remarks we derived based on the results of this study. We present evidence supporting the applicability of the method to new geological settings by making the arguments relevant to the results obtained from this study.

Please see the revised conclusions in the revised manuscript. |
| | **Technical corrections**

I refrain from making further technical corrections at this stage, but recommend that the authors consult a native English speaker for proofreading. | We sincerely thank the reviewer for a very careful and considerate examination of the manuscript and for providing us with very specific and detailed comments. We believe, revision following the reviewer's suggestions would |

| | I also recommend that the authors review the quality standards for submission to NHESS or any other journal and ensure that their manuscript meets these standards prior to submission. | definitely improve the quality of this manuscript to meet the standard of NHESS. We will surely ensure our revisions match your expectations prior to submission. |
|---|---|---|

**Review references**

Bogaard, T. and Greco, R.: Invited perspectives: Hydrological perspectives on precipitation intensity-duration thresholds for landslide initiation: proposing hydro-meteorological thresholds, Natural Hazards and Earth System Sciences, 18, 31–39, https://doi.org/10.5194/nhess-18-31-2018, 2018.

Brunetti, M. T., Peruccacci, S., Rossi, M., Luciani, S., Valigi, D., and Guzzetti, F.: Rainfall thresholds for the possible occurrence of landslides in Italy, Natural Hazards and Earth System Sciences, 10, 447–458, https://doi.org/10.5194/nhess-10-447-2010, 2010.

Destro, E., Marra, F., Nikolopoulos, E. I., Zoccatelli, D., Creutin, J. D., and Borga, M.: Spatial estimation of debris flows-triggering rainfall and its dependence on rainfall return period, Geomorphology, 278, 269–279, https://doi.org/10.1016/j.geomorph.2016.11.019, 2017.

D'Odorico, P., Fagherazzi, S., and Rigon, R.: Potential for landsliding: Dependence on hyetograph characteristics, Journal of Geophysical Research: Earth Surface, 110, https://doi.org/10.1029/2004JF000127, 2005.

Guzzetti, F., Peruccacci, S., Rossi, M., and Stark, C. P.: The rainfall intensity–duration control of shallow landslides and debris flows: an update, Landslides, 5, 3–17, https://doi.org/10.1007/s10346-007-0112-1, 2008.

Guzzetti, F., Gariano, S. L., Peruccacci, S., Brunetti, M. T., Marchesini, I., Rossi, M., and Melillo, M.: Geographical landslide early warning systems, Earth-Science Reviews, 200, 102973, https://doi.org/10.1016/j.earscirev.2019.102973, 2020.

Iadanza, C., Trigila, A., and Napolitano, F.: Identification and characterization of rainfall events responsible for triggering of debris flows and shallow landslides, Journal of Hydrology, 541, 230–245, https://doi.org/10.1016/j.jhydrol.2016.01.018, 2016.

Intrieri, E., Gigli, G., Casagli, N., and Nadim, F.: Brief communication: Landslide Early Warning System: toolbox and general concepts, Nat. Hazards Earth Syst. Sci., 13, 85–90, https://doi.org/10.5194/nhess-13-85-2013, 2013.

Mathew, J., Babu, D. G., Kundu, S., Kumar, K. V., and Pant, C. C.: Integrating intensity–duration-based rainfall threshold and antecedent rainfall-based probability estimate towards generating early warning for rainfall-induced landslides in parts of the Garhwal Himalaya, India, Landslides, 11, 575–588, https://doi.org/10.1007/s10346-013-0408-2, 2014.

Peruccacci, S., Brunetti, M. T., Gariano, S. L., Melillo, M., Rossi, M., and Guzzetti, F.: Rainfall thresholds for possible landslide occurrence in Italy, Geomorphology, 290, 39–57, https://doi.org/10.1016/j.geomorph.2017.03.031, 2017.

Scheevel, C. R., Baum, R. L., Mirus, B. B., and Smith, J. B.: Precipitation thresholds for landslide occurrence near Seattle, Mukilteo, and Everett, Washington, Precipitation thresholds for landslide occurrence near Seattle, Mukilteo, and Everett, Washington, U.S. Geological Survey, Reston, VA, https://doi.org/10.3133/ofr20171039, 2017.

Segoni, S., Rosi, A., Rossi, G., Catani, F., and Casagli, N.: Analysing the relationship between rainfalls and landslides to define a mosaic of triggering thresholds for regional-scale warning systems, Natural Hazards and Earth System Sciences, 14, 2637–2648, https://doi.org/10.5194/nhess-14-2637-2014, 2014.

Segoni, S., Piciullo, L., and Gariano, S. L.: A review of the recent literature on rainfall thresholds for landslide occurrence, Landslides, 15, 1483–1501, https://doi.org/10.1007/s10346-018-0966-4, 2018.

Stähli, M., Sättele, M., Huggel, C., McArdell, B. W., Lehmann, P., Van Herwijnen, A., Berne, A., Schleiss, M., Ferrari, A., Kos, A., Or, D., and Springman, S. M.: Monitoring and prediction in early warning systems for rapid mass movements, Natural Hazards and Earth System Sciences, 15, 905–917, https://doi.org/10.5194/nhess-15-905-2015, 2015.

Staley, D. M., Kean, J. W., Cannon, S. H., Schmidt, K. M., and Laber, J. L.: Objective definition of rainfall intensity–duration thresholds for the initiation of post-fire debris flows in southern California, Landslides, 10, 547–562, https://doi.org/10.1007/s10346-012-0341-9, 2013.

Thomas, M. A., Mirus, B. B., and Collins, B. D.: Identifying Physics-Based Thresholds for Rainfall-Induced Landsliding, Geophysical Research Letters, 45, 9651–9661, https://doi.org/10.1029/2018GL079662, 2018.

---

## Referee Report (RR1)

The manuscript under review is promising and caters to a very important issue, particularly, looking at the use of forecasted hourly rainfall data to simulate debris flows for early warning. This is also pertinent in areas where there is a dearth of data required to develop early warning systems, so a procedure that use synthetic data, but which is carefully validated is quite beneficial. Furthermore, this procedure can be useful in regions lacking historical rainfall data, which is rightly pointed out by the authors. The initial version of the manuscript indeed lacked relevant supporting literature, and the research objectives were not clearly articulated in the introduction; however, these shortcomings have been effectively addressed in the revised version. The Results and Discussion sections have been enhanced through an expanded presentation of the findings. Additionally, the manuscript commendably extends the application of rainfall intensity-duration thresholds, with a focus on Kedarnath, India, as observed in the revised version.

I note significant improvements in the figures, following recommendations from Referees 1 and 2. Nonetheless, I am confused by Figure 11. Despite captions indicating segments (a) through (i), only two sub-plots, labeled (a) and (b), are present, depicting varying cumulative rainfall intensities. This discrepancy suggests either an error in the figure, necessitating nine subsections, or a required revision of the caption. The previous version included nine subplots, which have been altered in the current version without updating the caption accordingly. Please update it accordingly.

Overall, the study exhibits robust scientific rigor, and the utilization of numerically synthesized data to model rainfall-threshold forecasts is commendable. However, I have some reservations regarding the numerical analysis and the resultant mapping of debris flow extents. Focusing on Figure 10 (a), while there is a notable correlation between the simulated debris flows of 2013 and actual events, numerous potential 'false positives' are apparent away from the river channel peripheries. It does prompt a question: could these be attributed to alternative debris flow mechanisms, such as those originating on hillslopes?

Despite these observations, the revisions undertaken, coupled with feedback from previous referees, indicate that the manuscript has successfully addressed key concerns. I would recommend to accept the manuscript, provided the suggested (small) revisions are made. I present a synopsis of the overall changes made by the reviewers:

**Methods Section**:- Detailed information about the analysis and calibration details for the debris flow model have been included, and this was important for reproducibility and validation of the methods used, addressing a significant gap in the original manuscript.

**Validation and Calibration of Models**:- The authors have now validated the Weather Research and Forecasting (WRF) model outputs with ground-based and satellite-derived precipitation data. They have also conducted a sensitivity analysis and used different empirical equations for debris flow volume estimation, which was a good step up from the previous single empirical equation based on the Taiwanese case.

**Overall Structure and Presentation**:- The authors have restructured the manuscript for better flow, clarity, and logical consistency. This includes adjusting the positioning of certain sections and adding necessary explanations and justifications for their modelling choices.

**Minor comments**:

Line 110: Although the referring to statistical thresholds holding physical explanations are true, it would be nice to see a reference citing/explaining this.

Line 230: In the examples of US, Italy, and Japan, please provide the relevant literature for reference.

Figure 12: In the legend, I do not see the year of publication for Lakhera et al. Are they referring to the same 2020 study that is mentioned in the caption?

---

## Author Response (AR2)

**NHESS**

**Ref: NHESS-2022-297**

**Title: Numerical model derived intensity-duration thresholds for early warning of rainfall-induced debris flows in a Himalayan catchment**

**Response to Editor**

| Ref. | Comment | Reply |
|------|---------|-------|
| 1 | Dear Authors,

Thank you very much for delicately incorporating all the suggestions by the first reviewers. The second round's reviewers think you have addressed all the issues, which I agree with. However, sharing the concern of the 1st reviewer, I believe some figures could be improved for clarity for the final publication.

Reviewer 1 has some minor concerns justifying some aspects of the manuscript and regarding Figures 11 and 12. Please improve the figures and the manuscript following the reviewer's suggestions. I also believe the text size (e.g., in the legend) in Figures 2, 10, and 11 is tiny to follow.

Once the figures meet NHESS standards, I will accept the article for publication.
Kind regards
Ugur Öztürk | Dear Editor Dr. Ugur Öztürk,

Thank you for valuable feedback and for acknowledging the revisions made in response to first-round reviewers' suggestions. We appreciate your consideration.

We have carefully addressed the concerns raised by you and the reviewer regarding Figures, ensuring that the improvement contributes to the clarity of the manuscript.

Additionally, we have also addressed the concerns about the text size in Figure 2,10, and 11 to comply with NHESS standards.

We hope these revisions meet the NHESS standards.

Kind regards.
Authors of NHESS-2022-297. |

**Response to Reviewers**

| Ref. | Comment | Reply |
|---|---|---|
| 1 | **General comments by Referee in Report #1**

The manuscript under review is promising and caters to a very important issue, particularly, looking at the use of forecasted hourly rainfall data to simulate debris flows for early warning. This is also pertinent in areas where there is a dearth of data required to develop early warning systems, so a procedure that use synthetic data, but which is carefully validated is quite beneficial. Furthermore, this procedure can be useful in regions lacking historical rainfall data, which is rightly pointed out by the authors. The initial version of the manuscript indeed lacked relevant supporting literature, and the research objectives were not clearly articulated in the introduction; however, these shortcomings have been effectively addressed in the revised version. The Results and Discussion sections have been enhanced through an expanded presentation of the findings. Additionally, the manuscript commendably extends the application of rainfall intensity-duration thresholds, with a focus on Kedarnath, India, as observed in the revised version. | Thank you for your careful consideration and very detailed evaluation of our manuscript.

We sincerely appreciate your time to provide constructive as well as critical comments on the manuscript, analysis, technical aspects and writing of this research article.

We thank you for your encouragement towards the intriguing idea of this manuscript and identifying it within the scientific scope of NHESS.

Please see our detailed responses below to each of your comments sectionized in the order. |
| 2 | **Review by Another Referee in Report #2** | Thank you for your careful consideration and evaluation of our manuscript. We sincerely thank the referee for having checklisted the manuscript excellent in scientific significance, good in scientific and presentation quality. |

Comments on Manuscript:

| Ref. | Comment | Reply |
|---|---|---|
| 1 | I note significant improvements in the figures, following recommendations from Referees 1 and 2.

Nonetheless, I am confused by Figure 11. Despite captions indicating segments (a) through (i), only two sub-plots, labeled (a) and (b), are present, depicting varying cumulative rainfall intensities. This discrepancy suggests either an error in the figure, necessitating nine subsections, or a required revision of the caption. The previous version included nine subplots, which have been altered in the current version without updating the caption accordingly. Please update it accordingly. | We deeply appreciate your thoughtful review and thorough evaluation of our manuscript.

Thank you for your careful observation and comment. We have updated the Figure 11 caption for better understanding. |
| 2 | Overall, the study exhibits robust scientific rigor, and the utilization of numerically synthesized data to model rainfall-threshold forecasts is commendable. However, I have some reservations regarding the numerical analysis and the resultant mapping of debris flow extents. Focusing on Figure 10 (a), while there is a notable correlation between the simulated debris flows of 2013 and actual events, numerous potential 'false positives' are apparent away from the river channel peripheries. It does prompt a question: could these be attributed to alternative debris flow mechanisms, such as those originating on hillslopes? | Thank you for your critical evaluation of the results and thoughtful questions. We think you are right. The model predicts the actual debris flow extents with an accuracy of 63 percent. However, the false positives are 37 percent. Those are spread over hillslopes away from river channels. We also believe the reason for this could be the different debris flow mechanisms which the model could not simulate satisfactorily.

Thank you again for this very important observation.

We explained the same in line 210 – 212 on page 13 in the revised manuscript. |
| 3 | Despite these observations, the revisions undertaken, coupled with feedback from previous referees, indicate that the manuscript has successfully addressed key concerns. I would recommend to accept the manuscript, provided the suggested (small) revisions are made. I present a synopsis of the overall changes made by the reviewers: | Thank you for your kind comment and in detail exploring the major revisions made to the manuscript. We hope to have addressed the minor comments made by you as stated in the above and below responses. |
| 4 | Methods Section:- Detailed information about the analysis and calibration details for the debris flow model have been included, and this was important for reproducibility and validation of the methods used, addressing a significant gap in the original manuscript. | Thank you for noting the crucial addition of detailed analysis and calibration information in the Methods section, addressing a significant gap for reproducibility. |
| 5 | Validation and Calibration of Models:- The authors have now validated the Weather Research and Forecasting (WRF) model outputs with ground-based and satellite-derived precipitation data. | Thank you for acknowledging our enhanced model validation, incorporating diverse precipitation data sources, and justifying the empirical approach for debris flow volume estimation. |

| | | |
|---|---|---|
| | They have also conducted a sensitivity analysis and used different empirical equations for debris flow volume estimation, which was a good step up from the previous single empirical equation based on the Taiwanese case. | |
| 6 | Overall Structure and Presentation:- The authors have restructured the manuscript for better flow, clarity, and logical consistency. This includes adjusting the positioning of certain sections and adding necessary explanations and justifications for their modelling choices. | Thank you for acknowledging the positive impact of our manuscript's restructuring on flow, clarity, and logical consistency. We remain committed to continuous improvement |

Minor Comments:

| Ref. | Comment | Reply |
|---|---|---|
| 1 | Line 110: Although the referring to statistical thresholds holding physical explanations are true, it would be nice to see a reference citing/explaining this. | Thanks for the suggestion. We have added the relevant reference for better understanding.

Please see page 11 line 162 in the revised manuscript. |
| 2 | Line 230: In the examples of US, Italy, and Japan, please provide the relevant literature for reference. | Thanks, we have included the references in page 15 Line 240 – 241 in the revised manuscript. |
| 3 | Figure 12: In the legend, I do not see the year of publication for Lakhera et al. Are they referring to the same 2020 study that is mentioned in the caption? | Thank you. Yes, we are referring to the same 2020 study mentioned in the caption. We have modified Figure 12 and included the year of publication. Please see revised Figure 12 on page 15 in the revised manuscript. |